# A synthetic metabolic network for physicochemical homeostasis

Tjeerd Pols[1,2], Hendrik R. Sikkema[1,2], Bauke F. Gaastra[1,2], Jacopo Frallicciardi[1], Wojciech M. Śmigiel [1], Shubham Singh[1] & Bert Poolman [1]

One of the grand challenges in chemistry is the construction of functional out-of-equilibrium networks, which are typical of living cells. Building such a system from molecular components requires control over the formation and degradation of the interacting chemicals and homeostasis of the internal physical-chemical conditions. The provision and consumption of ATP lies at the heart of this challenge. Here we report the in vitro construction of a pathway in vesicles for sustained ATP production that is maintained away from equilibrium by control of energy dissipation. We maintain a constant level of ATP with varying load on the system. The pathway enables us to control the transmembrane fluxes of osmolytes and to demonstrate basic physicochemical homeostasis. Our work demonstrates metabolic energy conservation and cell volume regulatory mechanisms in a cell-like system at a level of complexity minimally needed for life.

[1] Department of Biochemistry, Groningen Biomolecular Sciences and Biotechnology Institute & Zernike Institute for Advanced Materials, University of Groningen, Nijenborgh 4, 9747 AG Groningen, The Netherlands. [2]These authors contributed equally: Tjeerd Pols, Hendrik R. Sikkema, Bauke F. Gaastra. Correspondence and requests for materials should be addressed to B.P. (email: b.poolman@rug.nl)

The generation and consumption of ATP lies at the heart of life. Complex networks of proteins, nucleic acids and small molecules sustain the essential processes of gene expression and cell division that characterize living cells, but without ATP they are non-functional. Herein lies one of the major challenges in the construction of synthetic cell-like systems. Other processes, such as achieving tunable DNA replication, efficient transcription and translation, and vesicle division[1,2] are essentially secondary to the solution of a controlled energy supply. Metabolic energy conservation is a prerequisite for synthetic systems no matter how complex. Energy is critical not just for (macro)molecular syntheses but also for maintaining the cytoplasm in a state compatible with metabolism through control over pH, ionic strength and solute composition. Here we have addressed that issue and show that we can control ATP production and ionic homeostasis in synthetic vesicles.

The bottom-up construction of synthetic cells from molecular components[3] differs in concept and strategy from the top–down approach to engineer minimal cells, pioneered by the J. Craig Venter institute[4]. Yet, both approaches address what tasks a living cell should minimally perform and how this can be accomplished with a minimal set of components. New biochemical functions and regulatory principles will be discovered as we make progress toward constructing a minimal cell. In the field of bottom-up synthetic biology (perhaps better called synthetic biochemistry), work is progressing toward establishing new information storage systems[5], replication of DNA by self-encoded proteins[6], the engineering of gene and protein networks[7,8], formation of skeletal-like networks[9], biosynthesis of lipids[10–12], division of vesicles[13,14], development of non-lipid compartment systems[15,16] and chemical homeostasis through self-replication[17,18]. Protein synthesis has been realized using recombinant elements[19], which have been incorporated into vesicles[20,21] or water-in-oil droplets[16]. However, long-term sustained synthesis of chemicals is a bottleneck in the development and application of synthetic cell-like systems[22,23]. At the root of the poor performance of reconstituted systems are challenges that relate to sustained production of nucleotides, import of substrate(s) and export of waste product(s), control of the internal physicochemical conditions (pH, ionic strength, crowding) and stability of the lipid-bounded compartment, all of which require constant energy dissipation.

Inspired by the challenges of the bottom-up construction of a living cell, we focus on the development of new open vesicle systems that sustain nucleotide levels and electrochemical gradients to allow further functionalities to be integrated. ATP is especially crucial, not only as a source of metabolic energy for most biological processes, but also as a hydrotrope, influencing the viscosity and possibly the structure of the cytosol[24]. Energy consumption in a growing cell is dominated by polymerization reactions and maintenance processes[25], so regeneration of ATP is required to keep the cell running. Recent developments in the field of synthetic biochemistry have started to address the issue of ATP homeostasis. A cell-free molecular rheostat for control of ATP levels has been reported, employing two parallel pathways and regulation by free inorganic phosphate[26], but the system has not been implemented in vesicles. Photosynthetic artificial organelles have been constructed that form ATP on the outside of small vesicles, encapsulated in giant vesicles, allowing optical control of ATP dependent reactions[27,28].

Here, we present the construction of a molecular system integrated into a cell-like container with control of solute fluxes and tunable supply of energy to fuel ATP-requiring processes. We have equipped the vesicles with sensors for online readout of the internal ATP/ADP ratio and pH, allowing us to conclude that the system enables long-term metabolic energy conservation and physicochemical homeostasis.

## Results

**A system for sustained production of ATP.** The conversion of arginine into ornithine, ammonia plus carbon dioxide yields ATP in three enzymatic steps (Fig. 1a)[29]. For sustainable energy conservation in a compartmentalized system, the import of substrates and excretion of products have to be efficient, which can be achieved by coupling the solute fluxes. The antiporter ArcD2 facilitates the stoichiometric exchange of the substrate arginine for the product ornithine (Fig. 1b)[30], which is important for maintaining the metabolic network away from equilibrium. The thermodynamics of the arginine conversion under standard conditions are given in Fig. 1c. The equilibrium constant of the conversion of citrulline plus phosphate into ornithine plus carbamoyl-phosphate is highly unfavorable, but the overall standard Gibbs free energy difference ($\Delta G^0$) of the breakdown of arginine is negative. Since the actual $\Delta G$ is determined by $\Delta G^0$ and the concentration of the reactants, the antiport reaction favors an even more negative $\Delta G$ by maintaining an out-to-in gradient of arginine and in-to-out gradient of ornithine (Fig. 1d). We anticipate that $NH_3$ and $CO_2$ will passively diffuse out of the cell.

**Engineering of the metabolic network for ATP.** To construct the system for ATP regeneration, we purified and characterized arginine deiminase (ArcA), ornithine transcarbamoylase (ArcB), carbamate kinase (ArcC1), and the arginine/ornithine antiporter ArcD2. Their kinetic and molecular properties are summarized in Fig. 2a. The enzymatic network is enclosed with inorganic phosphate and Mg-ADP in vesicles composed of synthetic lipids, while ArcD2 is reconstituted in the membrane. The concentration and number of reporters, ions and metabolites per vesicle is given in Table 2. The lipid composition of the vesicles is based on general requirements for membrane transport (bilayer and non-bilayer-forming lipids, anionic and zwitterionic lipids), which is tuned to our needs (see below; ref. [31]). The vesicles obtained by extrusion through 400 and 200 nm filters have an average radius of 84 nm (SD = 59 nm; $n = 2090$) and 64 nm (SD = 39 nm; $n = 2092$), respectively (Fig. 2b and Supplementary Fig. 1), as estimated from cryo-electron microscopy images. The average internal volumes of the vesicles center around radii of 226 nm (SD = 113 nm) and 123 nm (SD = 49 nm), respectively (Fig. 2c). Although a fraction of the vesicles is multi-lamellar, it is likely that all layers of the vesicles are active because we reconstitute the membrane proteins in liposomes prior to the inclusion of the enzymes, sensors and metabolites (protocol A1). The encapsulation of the luminal components is done by five freeze-thaw cycles, which induce content exchange between vesicles and homogenize the membranes. The vesicles obtained by extrusion through 200 nm filters are more homogenous in size (Fig. 2b, c) but contain a smaller number of components (Fig. 2a and Table 2), yet the performance of the metabolic network is similar in both types of compartments (see below).

We first characterized ArcD2 in lipid vesicles without the enzymatic network and demonstrated exchange of arginine for ornithine (Fig. 2d). ArcD2 transports arginine in and ornithine out in both membrane orientations, which is a property of this type of secondary transporter. The direction of transport is determined by the concentration gradients of the amino acids, not by the orientation of the protein. The arginine/ornithine antiport reaction is not affected by an imposed membrane potential ($\Delta\Psi$) (Fig. 2d, inset). Surprisingly, we also detect that ArcD2 exchanges arginine for citrulline, albeit at a much lower rate than the arginine/ornithine antiport (Fig. 2e). The arginine/

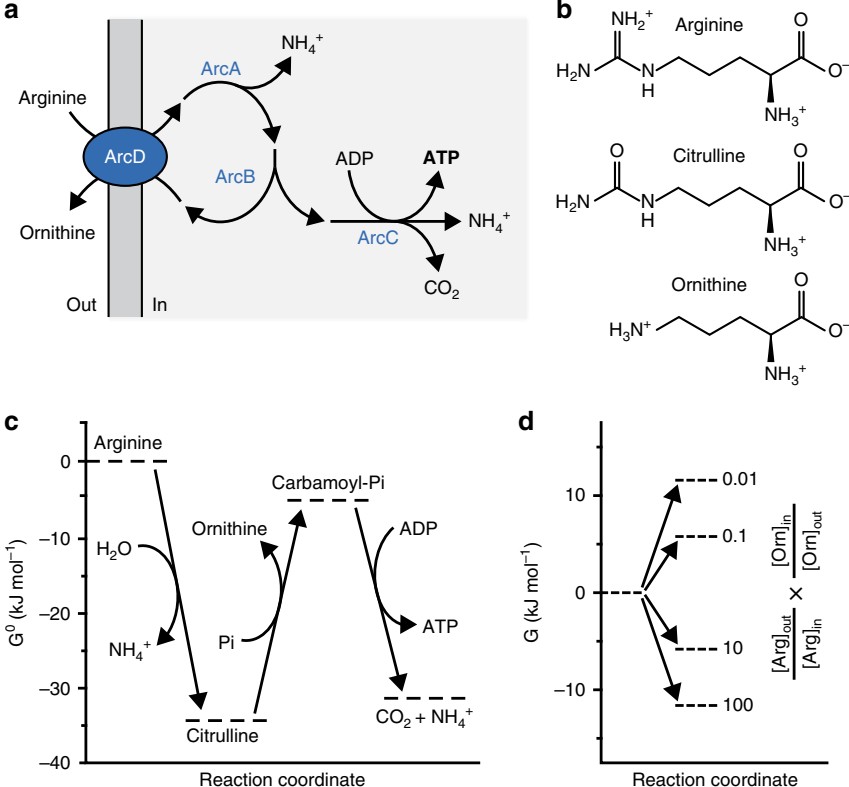

**Fig. 1** Layout and thermodynamics of the system. **a** Schematic representation of the arginine breakdown pathway. **b** Structures of arginine, citrulline and ornithine at neutral pH. **c** Thermodynamics of arginine conversion for the reactions of ArcA, ArcB, and ArcC1. $G^0$ values were calculated for pH 7.0 and an ionic strength of 0.1 M using eQuilibrator 2.2. **d** Thermodynamics of the arginine/ornithine antiport reaction. G values were calculated at varying concentration gradients of arginine (outside to inside) and ornithine (inside to outside). When the arginine and ornithine gradients are opposite, that is, $[Arg]_{in} < [Arg]_{out}$ and $[Orn]_{in} > [Orn]_{out}$, then a negative (and thus favorable) G value is obtained. Four scenarios for the product of the arginine and ornithine gradients are indicated

citrulline antiport is electrogenic (Fig. 2e, inset), which agrees with arginine (and ornithine) being cationic and citrulline being neutral at pH 7 (Fig. 1b).

The turnover number ($k_{cat}$) and equilibrium constant ($K_{eq}$) of the enzymes were used to guide the initial design of the pathway, and the enzymes were incorporated in the vesicles at a copy number well above the stochastic threshold (Fig. 2a). The ArcD2 protein is reconstituted at a lipid-to-protein ratio of 400:1 (w/w), yielding on average 62 and 19 antiporters per vesicle with radii of 226 and 123 nm, respectively. Since arginine is imported when a counter solute is present on the inside, we include L-ornithine in the vesicles to enable the metabolism of arginine. For readout of ATP production, we enclosed PercevalHR[32], a protein-based fluorescent reporter of the ATP/ADP ratio (Fig. 3a); the calibration and characterization of the sensor are shown in Supplementary Fig. 2. Upon addition of arginine, the vesicles produce ATP. Thus, after an initial, rapid, increase in the ratio of the excitation maxima at 500 nm and 430 nm (representing an increase in ATP/ADP ratio) the ratio unexpectedly declines after 30 min (Fig. 3b, blue trace). The ATP/ADP ratio increases again and stabilizes (Fig. 3b, black trace) in the presence of the protonophore carbonyl cyanide-4-(trifluoromethoxy) phenylhydrazone (FCCP), suggesting that arginine conversion by the metabolic network changes the internal pH of the vesicles (see below). The drop in fluorescence signal without FCCP is explained by the pH-dependent binding of nucleotides to PercevalHR (Supplementary Fig. 2b and 2c). The decrease in F500/430 signal suggests that the internal pH is decreasing,

because the fluorescence of PercevalHR increases with increasing pH (Supplementary Fig. 2b).

We found that, in the initial design of the pathway, a substantial amount of citrulline is formed on the outside, which is due to residual binding of ArcA to the outer membrane surface even after repeated washing of the vesicles; control experiments rule out that ArcA is binding to ArcD2 or OpuA or due to a high or low concentration of anionic lipid. We thus inactivated external ArcA by treatment of the vesicles with the membrane-impermeable sulfhydryl reagent *p*-chloromercuribenzene sulfonate (pCMBS). To avoid inhibition of the antiporter ArcD2, we engineered a cysteine-less variant that is fully functional and insensitive to sulfhydryl reagents (Supplementary Fig. 3). This optimized system is used for further characterization and application.

**Arginine breakdown and control of futile hydrolysis and pH.** The breakdown of arginine, in the lumen of the vesicle, is given by the reaction equation:

$$\text{L-Arginine} + H_2O + HPO_4^{2-} + \text{Mg-ADP}^{1-}$$
$$+ 3H^+ \rightarrow \text{L-Ornithine} + \text{Mg-ATP}^{2-} + 2NH_4^+ + CO_2 \quad (1)$$

The external pH increases upon the addition of arginine (Fig. 3c), which is in accordance with Eq. (1) when the reaction products (except for ATP) end up in the outside medium (Fig. 3d). Unexpectedly, the vesicle lumen acidifies over longer timescales, that is, after an initial increase of the internal pH

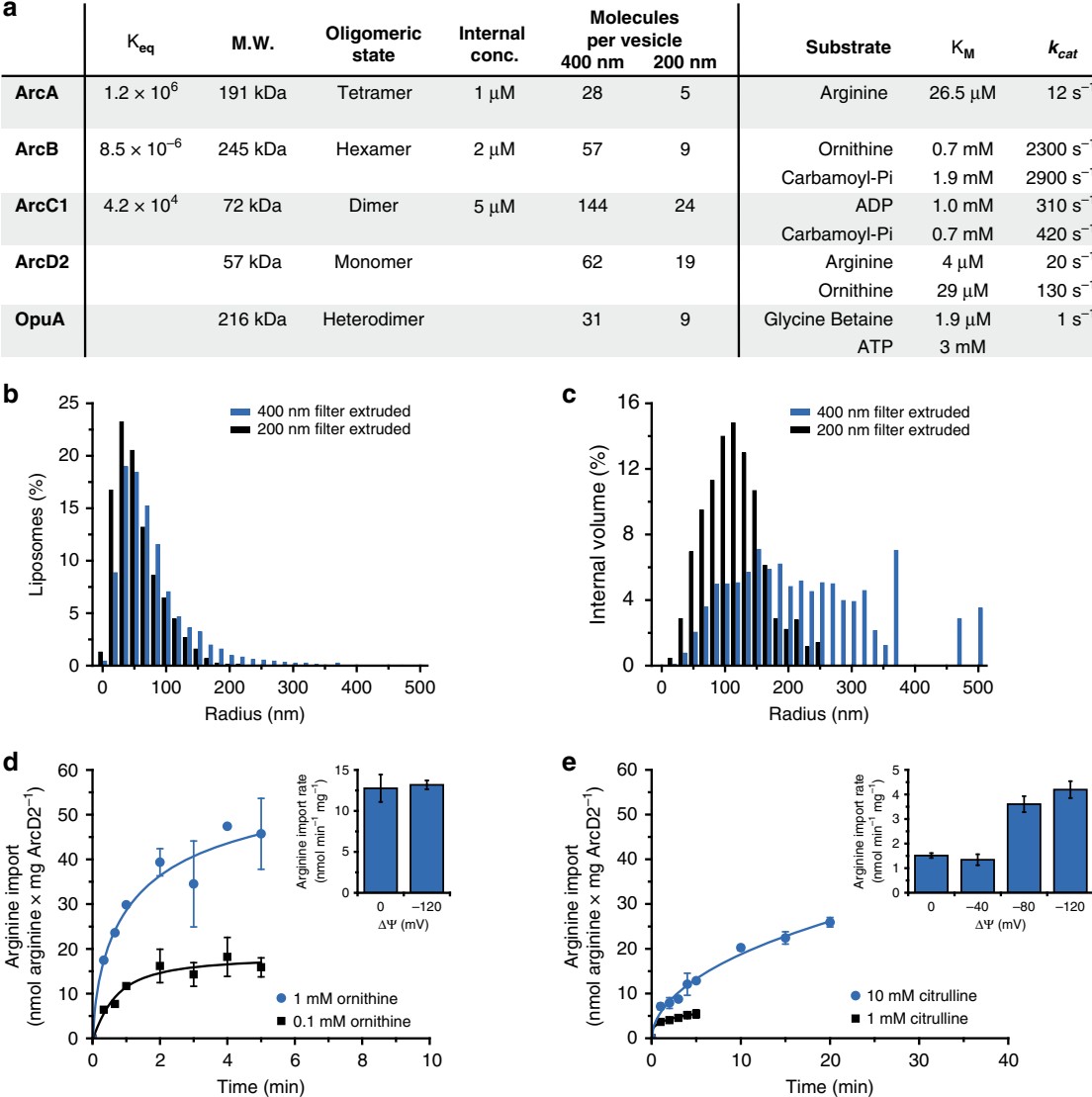

**Fig. 2** Characterization of the components of the system. **a** Molecular and kinetic properties of the enzymes; $K_{eq}$ was calculated as in Fig. 1c. The $K_M$ values of a given substrate were determined under conditions of excess of the other substrate. The number of molecules per vesicle was calculated from the internal concentration of the enzymes and the average size of the vesicles, for 400 nm (left column) and 200 nm (right column) extruded vesicles. Kinetic parameters of ArcB are given for the back reaction. The kinetic parameters of ArcD2 were estimated from measurements in cells[48], assuming that ArcD2 constitutes 1% of membrane protein; the data for OpuA are from ref. [40]. **b** Distribution of the radius of lipid vesicles extruded through a 400 nm (blue bars) and 200 nm (black bars) polycarbonate filter as estimated from CryoTEM micrographs (Supplementary Fig. 1). The diameter of 2090 vesicles (400 nm filter) and 2092 vesicles (200 nm filter) were measured using ImageJ. **c** Distribution of the internal volume, based on the distribution of radii, assuming that all vesicles are spherical. **d** Kinetics of arginine uptake in proteoliposomes with 1 mM (blue circles) and 0.1 mM (black squares) ornithine on the inside (protocol B2); $^{14}$C-arginine concentration of 10 μM. Inset: influence of a membrane potential (ΔΨ) on arginine uptake with 1 mM ornithine on the inside. Data from replicate experiments ($n = 2$) are shown, error bars indicate standard deviation. **e** Kinetics of arginine uptake in proteoliposomes with 10 mM (blue circles) and 1 mM (black squares) citrulline on the inside (protocol B2); $^{14}$C-arginine concentration of 10 μM. Inset: influence of a membrane potential (ΔΨ) on arginine uptake with 10 mM citrulline on the inside ($n = 2$). Source data of are provided as a Source Data file

(Fig. 2e, blue line; pyranine calibration shown in Supplementary Fig. 4); the transient in the internal pH is more evident when the internal buffer capacity is decreased (Fig. 2e, black line). The acidification of the vesicle lumen cannot be readily explained if arginine is solely converted into ornithine (Eq. (1)). Indeed, we found citrulline as an end product in addition to ornithine (Fig. 4a).

What is the basis for the futile hydrolysis of arginine? Since external ArcA was inactivated by pCMBS, we infer that part of the citrulline is not metabolized further but exported and

accompanied by diffusion of $NH_3$ through the membrane. This side reaction is possible if steps in the pathway downstream of ArcA are limiting the breakdown of arginine (Fig. 4b, bold arrow) and when citrulline is exchanged for arginine (Fig. 4b, dashed arrow). In the vesicles with the full arginine breakdown pathway, citrulline will compete with ornithine for export, when the internal citrulline concentration is high. The equilibrium constant of the reaction catalyzed by ArcB ($K_{eq} = 8.5 \times 10^{-6}$, see Fig. 2a) predicts high citrulline-to-ornithine ratios in the vesicles. Indeed, we find that the internal citrulline-to-ornithine ratio increases

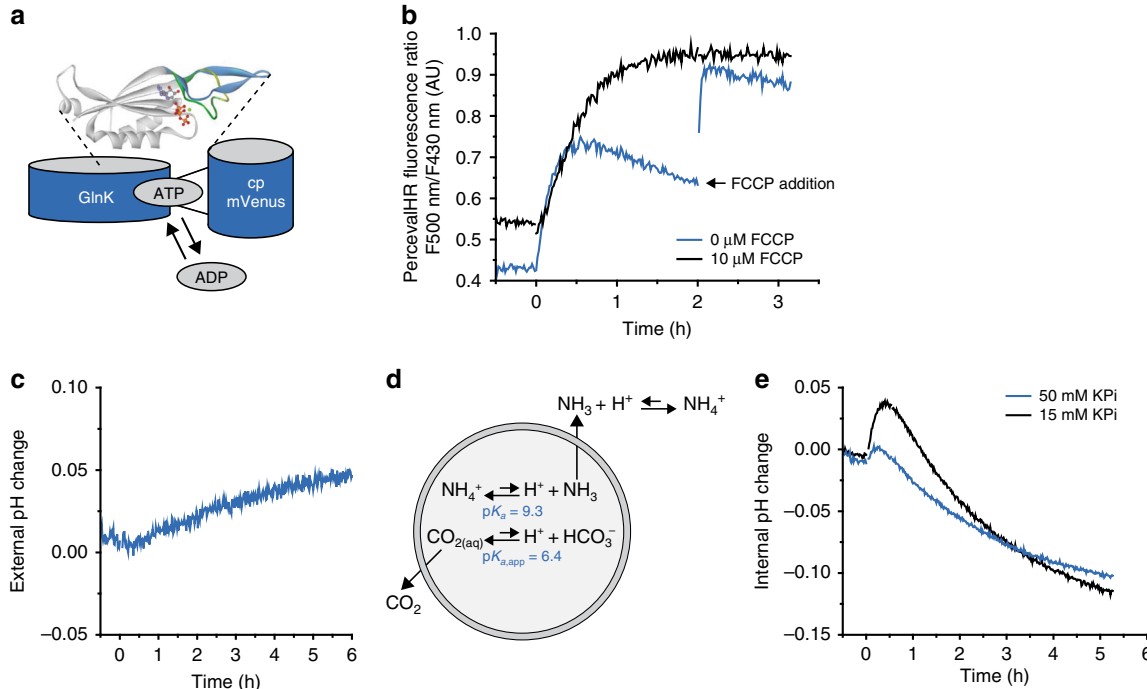

**Fig. 3** Generation of ATP and pH changes. The internal composition of the vesicles at the start of the experiment is given in Fig. 2a (enzymes) and Table 2 (ions, metabolites) and the preparation of vesicles is given in protocol A1, unless specified otherwise. **a** Schematic representation of PercevalHR; modification of cartoon in ref. [32]. **b** Effect of FCCP on the fluorescence readout of PercevalHR (protocol A1); the ratio of the fluorescence peaks at 500 nm and 430 nm is shown. In the absence of FCCP, the fluorescence readout declines 30 min after addition of 10 mM arginine (at $t = 0$) due to changes in the internal pH (addition of FCCP after 2 h increases the signal, indicated by the black arrow). The fluorescence signal is constant for several hours in the presence of 10 µM FCCP ($n = 2$). **c** External pH change (protocol B5) of arginine-metabolizing vesicles in outside medium with 10 mM KPi pH 7.0 plus 355 mM KCl. The ATP production was started by adding 5 mM arginine at $t = 0$. **d** Schematic representation of the pH effects caused by ammonia and carbon dioxide diffusion. **e** Internal pH change (protocol B4) of arginine-metabolizing vesicles with either 50 mM (blue trace) or 15 mM KPi plus 40 mM KCl, pH 7.0 on the inside (protocol A2; black trace). Five millimolar arginine was added at $t = 0$ ($n = 2$)

from 0 to more than 10 when arginine is converted for 1 h. Thus, the $K_{eq}$ values of the reactions of citrulline formation and breakdown (Fig. 2a), and the substrate promiscuity of ArcD2 (Fig. 2d, e) enable arginine/citrulline in addition to arginine/ornithine antiport.

How do the side reactions of the arginine breakdown pathway lead to acidification of the vesicle lumen? The $pK_A$ of $NH_4^+ \leftrightarrow NH_3 + H^+$ is 9.09 at 30 °C[33], and thus at pH 7.0 the fraction of $NH_3$ is small, but the base/conjugated acid reaction is fast. If $NH_3$ diffuses across the membrane, it will leave a proton behind in the vesicle lumen. Since the external volume is large compared with the internal one, there will be a net flux of $NH_3$ from the vesicle lumen to the medium. Using stopped-flow fluorescence-based flux measurements to probe the permeability of the vesicles for small molecules, we confirmed that $NH_3$ diffuses out rapidly, but that the membrane is highly impermeable for inorganic phosphate, $K^+$, and $Cl^-$ ions (Fig. 4c). $CO_2$ also diffuses rapidly across the membrane, down its concentration gradient, but only $NH_3$ leaves behind protons, therefore it is this that causes the pH change in the vesicle lumen. Finally, the breakdown of arginine to ornithine plus $NH_4^+$ and $CO_2$ is a dead-end process, which reaches equilibrium if the produced ATP is not utilized; the system runs out of ADP in about 30 min. The production of $NH_4^+$ from the conversion of arginine to citrulline then takes over, and the accompanying diffusion of $NH_3$ out of the vesicles leads to a net acidification of the vesicle lumen (Fig. 4d). Indeed, in Fig. 4d we show that the vesicles acidify significantly less when the vesicles are loaded with a higher concentration of ADP and the ATP synthesis is extended.

**Load on the metabolic network**. Cell growth is impacted by the solute concentration of the environment. Control of osmolyte import and export under conditions of osmotic stress allows cells to maintain their volume and achieve physicochemical homeostasis[34–36]. Potassium is the most abundant osmolyte in many (micro)organisms, but excessive salt accumulation increases the ionic strength, which diminishes enzyme function. To control the volume, internal pH, and ionic strength, bacteria modulate the intake of potassium ions. When needed, they replace the electrolyte for so-called compatible solutes, like glycine betaine, proline and/or sugars[37]. Compatible solutes like glycine betaine not only act in volume regulation but also prevent aggregation of macromolecules by affecting protein folding and stability[38,39].

The energy produced by the ATP breakdown pathway has been used to modulate the balance of osmolytes in the vesicles via activating the ATP-driven glycine betaine transporter OpuA (Fig. 4e). To this end we co-reconstituted OpuA with the components of the metabolic network for ATP production. OpuA transports solutes into the vesicle lumen when the protein is oriented with the nucleotide-binding domains on the inside. It happens to be that we reconstitute OpuA for more than 90% in this desired orientation[31,40], but any protein in the opposite orientation is non-functional because ATP is only produced inside the vesicles. In vesicles with 13 mol% DOPG [1,2-di-(9Z-octadecenoyl)-sn-glycero-3-phospho-(1′-rac-glycerol)] OpuA is constitutively active and imports glycine betaine at the expense of ATP (Fig. 5a), albeit at a low rate[41]. OpuA is ionic strength-gated and more active when sufficient levels of anionic lipids are present in the membrane[31,41], which is the physiologically more

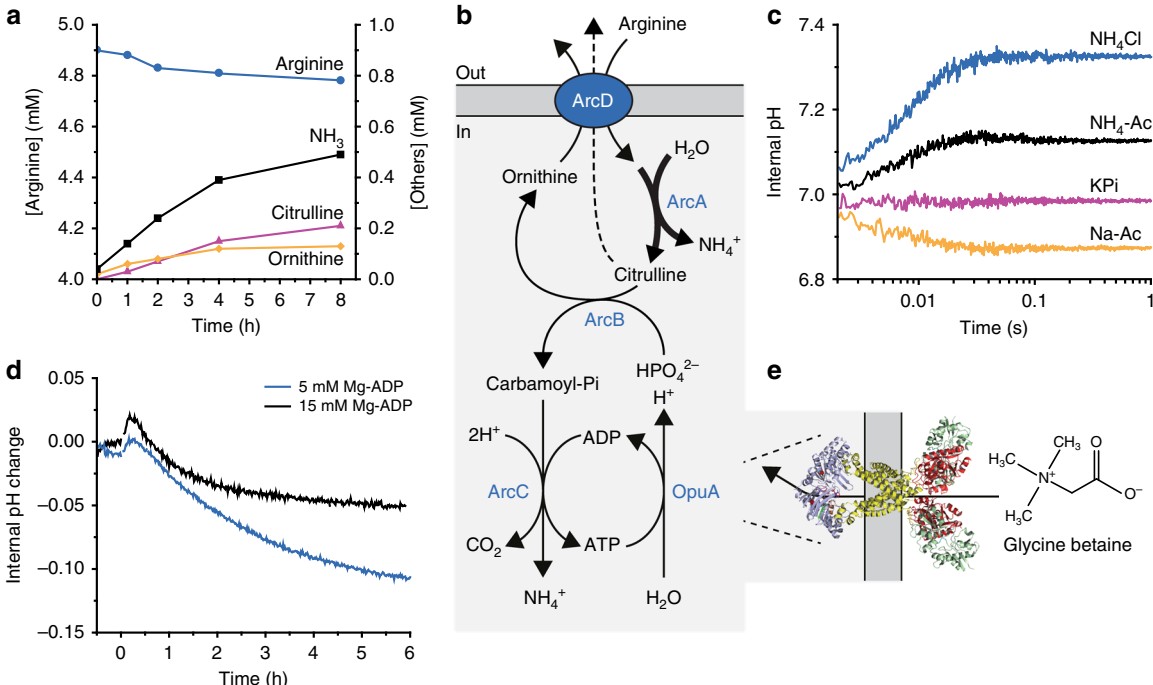

**Fig. 4** Control of futile hydrolysis of arginine and pH homeostasis. The internal composition of the vesicles at the start of the experiment is given in Fig. 2a (enzymes) and Table 2 (ions, metabolites) and the preparation of vesicles is given in protocol A1, unless specified otherwise. **a** External concentration of metabolites (protocol B6): arginine (blue circles), citrulline (pink triangles), ornithine (yellow diamonds), and $NH_3$ (black squares), as measured by HPLC in vesicles treated with 25 μM pCMBS. Five millimolar arginine was added at $t = 0$. **b** Schematic representation of arginine breakdown; the futile hydrolysis of arginine and arginine/citrulline exchange are depicted by bold and dashed arrows, respectively. **c** Stopped-flow fluorescence measurements to determine the permeability of the vesicles for $NH_4Cl$ (blue trace), $NH_4$-acetate (black trace), potassium phosphate (pink trace) and sodium acetate (yellow trace); pyranine inside the vesicles was used as pH indicator ($n \geq 2$). **d** Internal pH change (protocol B4) of arginine-metabolizing vesicles with either 5 mM Mg-ADP (protocol A1; blue trace) or 15 mM Mg-ADP on the inside (protocol A3; black trace). Five millimolar arginine was added at $t = 0$ ($n = 2$). **e** Homology model of OpuA and structure of the compatible solute glycine betaine. Glycine betaine import via OpuA consumes the ATP as indicated in (**b**)

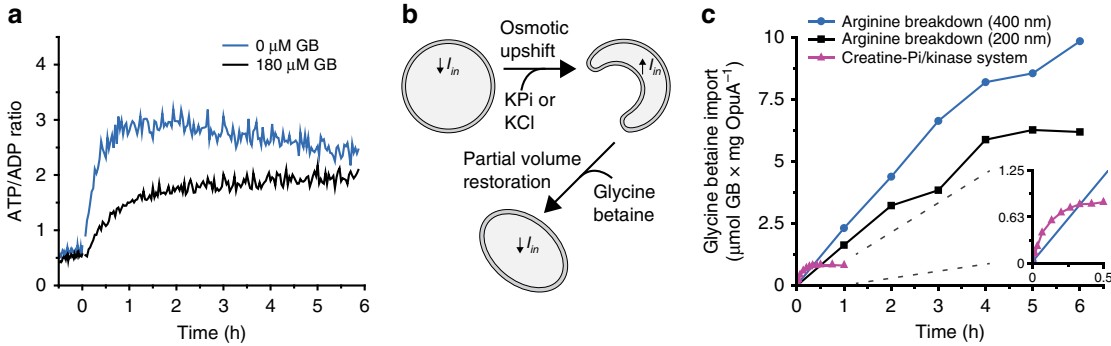

**Fig. 5** ATP/ADP homeostasis and long-term transport. The internal composition of the vesicles at the start of the experiment is given in Fig. 2a (enzymes) and Table 2 (ions, metabolites) and the preparation of vesicles is given in protocol A1, unless specified otherwise. **a** Effect of external glycine betaine (GB) on the ATP/ADP ratio measured by PercevalHR (protocol B3) in unshocked arginine-metabolizing vesicles (made with 13 mol% of DOPG; protocol A7), in the presence (black trace) and absence (blue trace) of 180 μM glycine betaine, added at $t = -0.5$ h. Five millimolar arginine was added at $t = 0$ ($n = 2$). **b** Schematic representation of the effect of osmotic upshift and partial volume restoration through glycine betaine uptake are shown. **c** Comparison of glycine betaine uptake (protocol B1) driven by ATP formed in the arginine breakdown pathway from 400 nm extruded vesicles (blue circles), 200 nm extruded vesicles (protocol A4; black squares), and the creatine-phosphate/kinase system (protocol A5; pink triangles) ($n = 2$)

relevant situation; for this, we use 38 mol% DOPG and an internal ionic strength below 0.2 M to lock OpuA in the off state. By increasing the medium osmolality with membrane-impermeant osmolytes (KPi or KCl), the vesicles shrink due to water efflux (Fig. 5b) until the internal osmotic balance is achieved, which occurs on the timescale of <1 s. Thus, the pressure exerted by the addition of KPi or KCl is dissipated by lowering the volume-to-surface ratio of the vesicles. The accompanying increase in internal ionic strength activates OpuA

and glycine betaine is imported to high levels (Fig. 5c, blue circles). The vesicles now possess an interior that is a mixture of glycine betaine and salts. The consumption of ATP by the gated import of glycine betaine is shown in Fig. 6a. Most remarkably, the gated import continues for hours when the internal ionic strength remains above the gating threshold and the pH is kept constant (Fig. 5c, blue circles). The open system, with arginine feed and product drain, performs at least an order of magnitude better than closed systems for ATP regeneration, where the

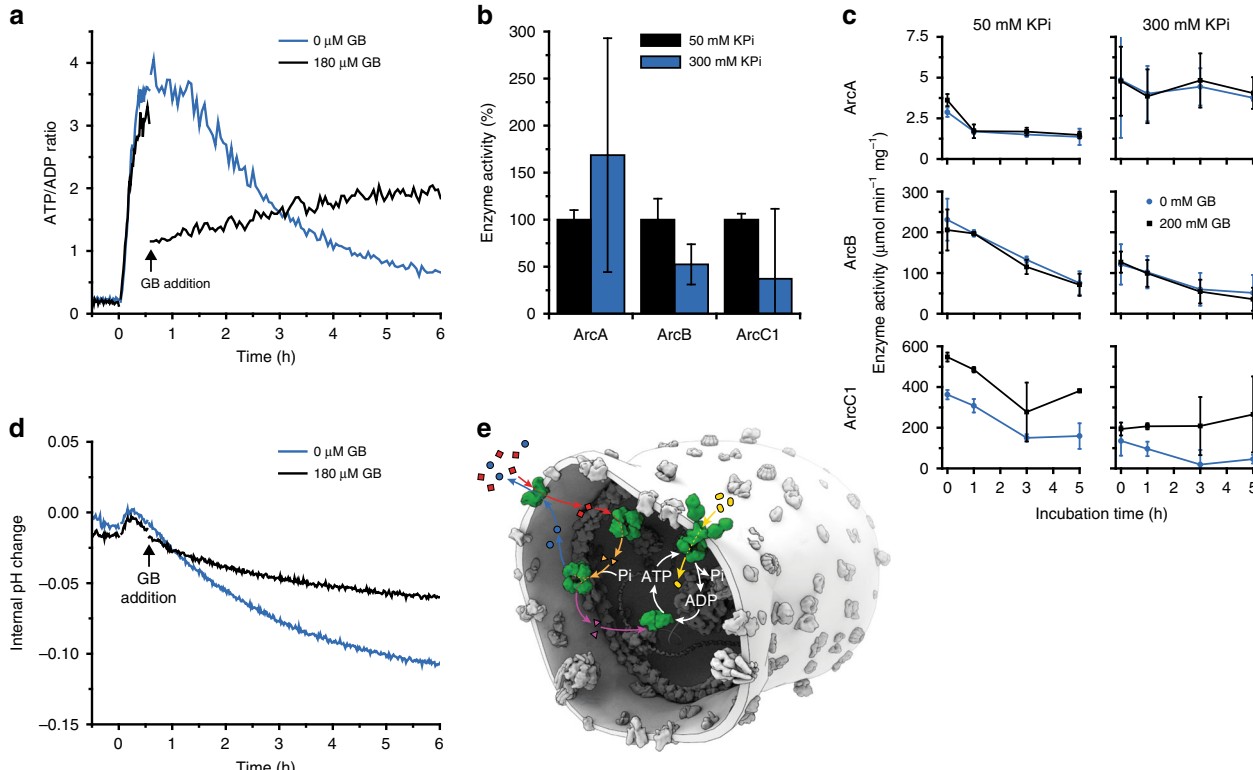

**Fig. 6** Physicochemical homeostasis of arginine-metabolizing vesicles. The internal composition of the vesicles at the start of the experiment is given in Fig. 2a (enzymes) and Table 2 (ions, metabolites) and the preparation of vesicles is given in protocol A1, unless specified otherwise. **a** Effect of glycine betaine (GB) on the ATP/ADP ratio measured by PercevalHR (protocol B3) inside arginine-metabolizing vesicles exposed to an osmotic upshift (addition of 250 mM KCl externally) in the presence (black trace) and absence (blue trace) of 180 μM glycine betaine, added at $t = 0.5$ h. Five millimolar arginine was added at $t = 0$ ($n = 3$). **b** Activity of ArcA (left), ArcB (middle), and ArcC1 (right) in 50 mM KPi, pH 7.0 (black bar), and 300 mM KPi, pH 7.0 (blue bar) as determined from the production of citrulline (ArcA, ArcB) or ATP (ArcC1). The activities were normalized to those in 50 mM KPi, pH 7.0, the absolute activities are given in Supplementary Table 3; error bars indicate standard deviation ($n = 2$). **c** Stability of ArcA (top), ArcB (middle), and ArcC1 (bottom) in 50 mM KPi, pH 7.0 (left), and 300 mM KPi, pH 7.0 (right) at 30 °C, in the presence (black squares) and absence (blue circles) of 200 mM glycine betaine ($n = 2$). **d** Effect of glycine betaine on the internal pH measured by pyranine (protocol B4) inside arginine-metabolizing vesicles exposed to an osmotic upshift (250 mM KCl) in the presence (black trace) and absence (blue trace) of 180 μM glycine betaine, added at $t = 0.5$ h. Five millimolar arginine was added at $t = 0$ ($n = 3$). **e** Schematic representation of the synthetic metabolic network in the cell-like container with the intermediates: arginine (red squares), ornithine (blue circles), citrulline (orange triangles), carbamoyl-phosphate (pink triangles), and glycine betaine (yellow ovals). $NH_3$ and $CO_2$ are not shown. Source data of are provided as a Source Data file

substrate for ATP synthesis is present on the inside and cannot be replenished, as exemplified by the creatine-phosphate/kinase system (Fig. 5c, pink triangles)[40]. Comparable results were obtained when smaller, yet more homogenous vesicles were formed by extrusion through 200 nm polycarbonate filters (Fig. 5c, black squares, see also Supplementary Fig. 5).

To determine the fraction of vesicles with a fully functional arginine breakdown pathway, we compared the rates of transport of glycine betaine in our synthetic cell system with those of OpuA vesicles containing 10 mM of ATP. With our synthetic network transport only occurs when ATP is formed, which requires the presence of each of the enzymes well above the stochastic threshold. When the metabolic pathway reaches steady state, the ATP and ADP concentrations are about 3.3 and 1.7 mM, respectively (see Table 2, ATP/ADP ratio of 2), and under these conditions we determined the rate of transport via OpuA. The $K_M$ value for ATP is 3 mM and the $K_I$ for ADP is 12 mM (taken from ref. [42]). From these numbers we compute the $V/V_{MAX}$ for transport in the synthetic vesicles. For the OpuA vesicles with 10 mM ATP there is negligible formation of ADP when initial rates of transport are determined, and the $V/V_{MAX}$ is calculated similarly. From the ratio of the $V/V_{MAX}$ in the vesicles with full

pathway over the $V/V_{MAX}$ in the OpuA vesicles (Fig. 5c), we obtain a lower limit for the fraction of active vesicles of 70%.

**Physicochemical homeostasis.** Next, we tested how the physicochemical homeostasis is sustained when the vesicles are osmotically challenged. Figure 6a shows the evolution in time of the ATP/ADP ratio upon addition of arginine to vesicles that were exposed to an increased medium osmolality 30 min before the addition of arginine; FCCP was added to avoid pH effects on the readout of PercevalHR (Supplementary Fig. 2b). In the absence of glycine betaine the ATP/ADP ratio peaks at 1 h and decreases over the next 4–5 h (Fig. 6a, blue line), which indicates the presence of futile ATP hydrolysis when the internal salt concentration is high. Intriguingly, when glycine betaine was added 0.5 h after arginine (Fig. 6a, black line), the ATP/ADP ratio drops instantly, but then remains stable for 6 h. Glycine betaine therefore has two effects: its accumulation provides a 'cytosol' that is more compatible with enzyme function (Fig. 6b, c), but it also provides a metabolic sink for ATP through its OpuA-mediated transport. The decrease in ATP/ADP ratio in the absence of glycine betaine can be explained by inhibition of enzymatic activity at high ionic strength (Fig. 6b), combined with

futile hydrolysis of ATP. Accordingly, the decrease in ATP/ADP ratio is much less in vesicles in which the ionic strength is kept low (Fig. 5a); compare Figs. 5a and 6a. Finally, we report ATP/ADP ratios because the absolute concentrations change when the vesicles are osmotically shrunk and subsequently regain volume. In most experiments, the initial adenine nucleotide (= ADP) concentration was 5 mM but increases when the vesicles are exposed to osmotic stress. In Fig. 5a, we report data of unshocked vesicles and here the ATP/ADP ratios of 2–3 correspond to 3.33–3.75 mM of ATP, respectively, which is in the range of concentrations in living cells.

Importantly, the introduction of an ATP-consuming process stimulates the full pathway at the expense of citrulline formation, and hence should stabilize the internal pH. Indeed, our results show that the system is better capable of maintaining the internal pH relatively constant when ATP is utilized (Fig. 6d, black line). Under these conditions the arginine-to-citrulline conversion is diminished relative to full pathway activity. We therefore propose that the stabilizing effect of glycine betaine accumulation originates from a lowering of the ionic strength (partial restoration of the vesicle volume), a chaperoning effect on the proteins and the maintenance of the internal pH, hence reflecting what compatible solutes do in living cells[37].

In summary and perspective: we present the in vitro construction of a cell-like system that maintains a metabolic state far-from-equilibrium for many hours (Fig. 6e). This is one of the most advanced functional reconstitutions of a chemically defined network ever achieved, which allows the development of complex life-like systems with adaptive behavior in terms of lipid and protein synthesis, cell growth and intercellular communication. We show that ATP is used to fuel the gated transport of glycine betaine, which allows the synthetic vesicles to maintain a basic level of physicochemical homeostasis. We have recently postulated alternative mechanisms for metabolic energy conservation[43] by coupling substrate/product antiporters to substrate decarboxylation, allowing the formation of a proton or sodium motive force. Combining such a pathway with the here-developed network for ATP would allow an even greater control of the physicochemical homeostasis.

Maintenance of the ATP/ADP ratio, internal pH, and presumably ionic strength are crucial for any metabolic system in the emerging field of synthetic biochemistry[23,26]. We expect that our metabolic network will find wide use beyond membrane and synthetic biology, as biomolecular out-of-equilibrium systems will impact the development of next generation materials (e.g. delivery systems) with active, adaptive, autonomous, and intelligent behavior.

## Methods

**Materials.** Common chemicals were of analytical grade and ordered from Sigma-Aldrich Corporation, Carl Roth GmbH & Co. KG or Merck KGaA. The lipids were obtained from Avanti Polar Lipids, Inc. (> 99% pure, in chloroform): 1,2-dioleoyl-*sn*-glycero-3-phosphoethanolamine (DOPE) [850725C], 1,2-dioleoyl-*sn*-glycero-3-phosphocholine (DOPC) [850375C], 1,2-dioleoyl-*sn*-glycero-3-phospho-(1′-rac-glycerol) (DOPG) [840475C]. *n*-dodecyl-β-D-maltoside (DDM) [D97002] was purchased from Glycon Biochemicals GmbH and Triton X-100 [T9284] from Sigma-Aldrich Corporation. ¹⁴C-glycine betaine was prepared enzymatically from ¹⁴C-choline-chloride (American Radiolabeled Chemicals, Inc. [ARC 0208, 55 mCi mmol⁻¹]) as described in ref. [31]. ¹⁴C-arginine was purchased from Moravek, Inc. [MC-137, 338 mCi mmol⁻¹], ¹⁴C-citrulline from American Radiolabeled Chemicals, Inc. [ARC 0508, 55 mCi mmol⁻¹], and ³H-ornithine hydrochloride from PerkinElmer Health Sciences, Inc. [NET1212, 21.4 Ci mmol⁻¹].

**Construction of expression strains.** The *arcX* genes were PCR-amplified from the genome of *Lactococcus lactis* IL1403 with primers *arcX*-Fw and *arcX*-rev (see Supplementary Table 1), using Phusion HF DNA polymerase (Thermo Fisher Scientific, Inc.). The *arcA* and *arcB* PCR inserts, and the pNZcLICoppA vector[44], were digested with NcoI and BamHI and subsequently ligated to yield pNZ*arcA* and pNZ*arcB*. These vectors contain the corresponding genes under the control of

**Table 1 Buffers used in this study**

| Buffer | Composition |
|---|---|
| A | 100 mM KPi, pH 7.0 |
| B | 50 mM KPi, pH 7.0 |
| C | 50 mM KPi, pH 7.0 with 200 mM NaCl |
| D | 50 mM KPi, pH 7.0 with 100 mM NaCl |
| E | 25 mM KPi, pH 8.0 with 500 mM NaCl plus 5% (v/v) glycerol |
| F | 50 mM KPi, pH 7.0 with 200 mM KCl |
| G | 100 mM KPi, pH 7.0 with 0.5 mM ornithine |
| H | 50 mM NaPi, pH 7.0 |
| I | 100 mM KPi, pH 7.0 with 250 mM KCl |

the nisin-inducible $P_{NIS}$ promoter[45] and the genes have a cleavable 6His-tag at the N-terminus.

The *arcC1* and *arcD2* PCR inserts were used for ligation-independent cloning as described in ref. [46]. This yielded pNZ*arcC1* and pNZ*arcD2* with the genes under the control of the $P_{NIS}$ promoter and with a cleavable 10His-tag at the N- and C-terminus, respectively. To construct the cysteine-less variant of *arcD2* (*arcD2*ΔC), two mutations were made, namely C395T and C487T. The *arcD2* gene was PCR-amplified from pNZ*arcD2* with uracil-containing primers (*arcD2*ΔC-X), using PfuX7 DNA polymerase[47]. The two amplified fragments were ligated with USER enzyme (New England Biolabs, Inc.) to create the pNZ*arcD2*ΔC vector.

The pNZ*arcA*, pNZ*arcB*, and pNZ*arcC1* vectors were transformed into *L. lactis* NZ9000[45], while pNZ*arcD2* and pNZ*arcD2*ΔC were transformed into *L. lactis* JP9000 Δ*arcD1D2*[48] (Supplementary Table 2). The pRsetB-PercevalHR plasmid was a gift from professor Gary Yellen (Addgene plasmid #49081, ref. [32]) and transformed into *E. coli* BL21-DE3 (Supplementary Table 2).

**Expression of genes.** *L. lactis* cells were grown in rich medium [2% (w/v) Gistex from Brenntag AG], 65 mM NaPi, pH 7.0, 1% (w/v) glucose) with 5 µg mL⁻¹ chloramphenicol at 30 °C with stirring (200 rpm). The strains for ArcA, ArcB, and ArcC1 production were grown as 3 L cultures in 5 L flasks and induced at an optical density at 600 nm (OD₆₀₀) of 0.5 with 0.05% (v/v) of culture supernatant from a nisin A-producing strain[45]. In contrast, the strains for ArcD2 and OpuA were grown as 2 L cultures in a 3 L bioreactor stirred at 200 rpm with pH control (kept above pH 6.5 with 4 M KOH) and induced at an OD₆₀₀ of 2.0 with 0.05% (v/v) of culture supernatant from a nisin A-producing strain. After induction, all strains were grown for an additional 2 h before harvesting. Harvesting and washing was done by centrifugation (15 min, 6000 × *g*, 4 °C) and resuspension of the cells in ice-cold 100 mM KPi, pH 7.0 (buffer A, Table 1). Finally, cells were centrifuged again and resuspended to an OD₆₀₀ of 100 in ice-cold 50 mM KPi, pH 7.0 (buffer B), flash-frozen in liquid nitrogen in aliquots of 50 mL and stored at −80 °C.

**Preparation of cell lysates and membrane vesicles.** The preparation of cell lysate and membrane vesicles was done as follows. After cells were thawed on ice, 100 µg mL⁻¹ DNAse and 2 mM MgSO₄ were added. Cells were lysed by high-pressure disruption (Constant Systems, Ltd.) with two passages at 39 kpsi and 4 °C. After lysis, 5 mM Na₂-EDTA (pH 8.0) and 1 mM PMSF (100 mM stock in iso-propanol) were added and cell debris was removed by centrifugation (15 min, 22,000 × *g*, 4 °C). Next, the supernatant was centrifuged for 90 min at 125,000 × *g* at 4 °C. For ArcA, ArcB, and ArcC1, the supernatant (containing cell lysate) was flash-frozen in liquid nitrogen in 10 mL aliquots and stored at −80 °C. For ArcD2 and OpuA, the pellets (containing membrane vesicles) were resuspended in ice-cold buffer B [with 20% (v/v) glycerol for OpuA], to a 10 mg mL⁻¹ protein concentration before flash-freezing (2 mL aliquots) and storage.

*E. coli* BL21-DE3 pRsetB-PercHR cells were grown in lysogeny broth (LB) with 100 µg mL⁻¹ ampicillin. In total, 0.5 L cultures were grown in 5 L flasks at 30 °C with shaking (200 rpm) to an OD₆₀₀ of 0.7, after which they were cooled to 22 °C, induced with 100 µM isopropyl β-D-1-thiogalactopyranoside (IPTG) and grown for an additional 72 h before harvesting as described in ref. [32]. Harvesting and washing was done by centrifugation (15 min, 6000 × *g*, 4 °C) and resuspension in ice-cold 50 mM NaPi, pH 7.0 with 100 mM NaCl. Finally, cells were centrifuged again and resuspended to an OD₆₀₀ of 100 in ice-cold 20 mM NaPi, pH 7.0 with 100 mM NaCl, flash-frozen in liquid nitrogen in 50 mL aliquots and stored at −80 °C. Cell lysate was prepared by adding 250 µg of DNAse to thawed cells, after which they were lysed by high-pressure disruption in a single passage at 25 kpsi and 4 °C. After lysis, 0.1 mM PMSF was added and cell debris was removed by centrifugation for 60 min at 145,000 × *g* at 4 °C. The supernatant (containing cell lysate) was flash-frozen in liquid nitrogen in 15 mL aliquots and stored at −80 °C.

**Purification of ArcA, ArcB, and ArcC1.** All protein purification and handling steps were performed on ice or at 4 °C, except when specified otherwise. Ni²⁺-Sepharose resin was pre-equilibrated in 50 mM KPi, pH 7.0, with 200 mM NaCl (buffer C) with either 10 mM imidazole for ArcA and ArcB or 5 mM imidazole and 10% (v/v) glycerol for ArcC1. Cell lysate was thawed on ice, added to the Ni²⁺-Sepharose

resin (0.5 mL bed volume per 10 mg total protein) and nutated for 1 h. The mixture was poured over a poly-prep column (Bio-Rad Laboratories, Inc.), after which the resin was washed with 20 column volumes of buffer C with 50 mM imidazole [plus 10% (v/v) glycerol for ArcC1]. Proteins were then eluted with three column volumes of buffer C with 500 mM imidazole [plus 10% (v/v) glycerol for ArcC1]. The most concentrated fractions were run on a Superdex 200 Increase 10/300 GL size-exclusion column (GE Healthcare) in 50 mM KPi, pH 7.0 with 100 mM NaCl (buffer D) [plus 10% (v/v) glycerol for ArcC1]. Protein containing fractions were pooled and concentrated to 4–5 mg mL$^{-1}$ in a Vivaspin 500 (30,000 kDa) centrifugal concentrator (Sartorius AG), after which they were aliquoted, flash-frozen in 100 µL aliquots and stored at −80 °C.

In addition, ArcA, ArcB, and ArcC1 have been purified in 50 mM NaPi, pH 7.0 instead of 50 mM KPi pH 7.0 to allow reconstitutions devoid of potassium ions (protocol A6, see below); the system is also fully functional in sodium ion-based buffers.

**Purification of PercevalHR.** PercevalHR was purified in a manner similar to ArcA, ArcB, and ArcC1, except that different buffers were used. Ni$^{2+}$-Sepharose resin was pre-equilibrated in 25 mM KPi buffer, pH 8.0 with 500 mM NaCl with 5% (v/v) glycerol (buffer E) plus 10 mM imidazole. The resin was washed with buffer E plus 25 mM imidazole and protein was eluted with buffer E plus 250 mM imidazole. The most concentrated fractions were run on a Superdex 200 Increase 10/300 GL size-exclusion column (GE Healthcare) in 10 mM NaPi, pH 7.4 with 150 mM NaCl and 5% (v/v) glycerol. Protein containing fractions (1–2 mg mL$^{-1}$) were aliquoted in volumes of 50 µL, flash-frozen and stored at −80 °C.

**Enzymatic assays for ArcA and ArcB.** Activity of ArcA and ArcB was measured with the COLDER assay[49]. In brief, either 2 µg mL$^{-1}$ ArcA, or 0.25 µg mL$^{-1}$ ArcB was incubated in buffer B at 30 °C for 3 min, in a total volume of 275 µL. To start the reaction, varying concentrations of either L-arginine for ArcA (0–480 µM L-arginine), or L-ornithine plus carbamoyl-phosphate for ArcB (0–10 mM L-ornithine plus 0–10 mM carbamoyl-phosphate) were added. Two hundred microliters of COLDER solution (20 mM 2,3-butanedione monoxime, 0.5 mM thiosemicarbazide, 2.25 M phosphoric acid, 4.5 M sulfuric acid and 1.5 mM ammonium iron(III) sulfate) was pipetted into each well of a 96-well flat-bottom transparent polystyrene plate (Greiner Bio-One International GmbH), to which 50 µL of reaction mixture was added at given time intervals to stop the enzymatic conversion. In addition, a set of calibration samples with citrulline concentrations from 0 to 250 µM was added into the wells in the plate. To allow color development, the plate was sealed with thermo resistant tape (Nalge Nunc International) and incubated at 80 °C for 20 min in a block heater (Stuart). Afterward, the plate was cooled down to room temperature for 30 min, the condensate was centrifuged (1 min, 1000 × g, 20 °C) and the absorbance of the solutions in the wells was measured at 540 nm in a platereader (BioTek Instruments, Inc.). Enzyme activity (in nmol citrulline min$^{-1}$ mg protein$^{-1}$) was determined by the formula:

$$\text{Act}_{enz} = \frac{\Delta \text{enz}}{\Delta \text{cal}} * \frac{1}{C_{enz} * \text{Vol}_{rmx}} \quad (2)$$

where Δenz and Δcal are the slopes of the enzyme and calibration curves, respectively, in AU min$^{-1}$ and AU nmol$^{-1}$ citrulline; C$_{enz}$ is the final concentration of enzyme in mg mL$^{-1}$ and Vol$_{rmx}$ is the volume of the reaction mixture in mL.

Stability measurements of ArcA and ArcB were performed as described above, with minor adjustments. In brief, ArcA and ArcB were diluted in either buffer B or 300 mM KPi, pH 7.0, with and without 200 mM glycine betaine, and incubated at 30 °C for 0, 1, 3, and 5 h. To start the reaction, either 150 µM arginine (for ArcA) or 5 mM carbamoyl-phosphate plus 5 mM citrulline (for ArcB) were added.

**Enzymatic assays for ArcC1.** The activity of ArcC1 was measured from changes in ATP/ADP ratio with PercevalHR. In all, 3.3 µg mL$^{-1}$ ArcC1 was incubated in buffer B supplemented with 5 mM of MgSO$_4$ and 10 µg mL$^{-1}$ of purified PercevalHR in 105.250-QS cuvettes (Hellma Analytics) in a FP-8300 spectrofluorometer (Jasco, Inc.). ADP was added in varying concentrations (0.1–10 mM) and the mixture was incubated at 30 °C for 5 min. To start the reaction, carbamoyl-phosphate was added in varying concentrations (0.2–10 mM). The fluorescence spectrum of PercevalHR was measured by excitation from 400 ± 5 to 510 ± 5 nm, while the emission was recorded at 550 ± 5 nm. As the pH of the reaction mixture changes in time and PercevalHR is sensitive to pH, it was necessary to measure the pH changes and correct the PercevalHR readout accordingly. In the pH experiments PercevalHR was substituted with 0.1 µM pyranine and the fluorescence spectrum of pyranine was measured by excitation from 380 ± 5 to 480 ± 5 nm, while the emission was recorded at 512 ± 5 nm. Pyranine was calibrated as described under internal pH measurements with pyranine (see below).

The PercevalHR signal was calibrated at a sensor concentration of 10 µg mL$^{-1}$ in buffer B with varying pH values (from 6.6 to 7.6), containing a mixture of ATP and ADP at a total concentration of 5 mM and a total MgSO$_4$ concentration of 5 mM. The ATP/ADP ratio was plotted against the ratio of the two excitation peaks

at 430 and 500 nm and were fitted using the Hill equation:

$$\frac{\text{F500nm}}{\text{F430nm}} = \text{start} + (\text{end} - \text{start}) \frac{\left(\frac{\text{ATP}}{\text{ADP}}\right)^n}{k^n + \left(\frac{\text{ATP}}{\text{ADP}}\right)^n} \quad (3)$$

where $k$ is the apparent affinity constant; $n$ is the Hill coefficient; start and end refer to the y-values at the vertical asymptotes. The Hill coefficient was constrained to 1. When the parameters of datasets recorded at varying pH values were compared, it was evident that only start and end were affected by pH; $k$ remained constant. The start and end values were plotted against the pH and were fitted by using the following logistic equations (Supplementary Fig. 2c):

$$\text{start} = y_0 + A \times e^{R_0 \times \text{pH}}, \quad \text{end} = y_0 + A \times e^{R_0 \times \text{pH}} \quad (4)$$

where $y_0$, $A$, and $R_0$ are the fit parameters. The resulting equations were incorporated into Eq. (3) to yield a formula in which the ATP/ADP ratio is dependent on the pH of the solution and the ratio of the excitation peaks at 430 and 500 nm of PercevalHR:

$$\frac{\text{ATP}}{\text{ADP}} = \frac{4.11 \times \left(\frac{\text{F500nm}}{\text{F430nm}} - \left(-0.021 + 2 \times 10^{-6} \times e^{1.76 \times \text{pH}}\right)\right)}{\left(-1.15 + 3.5 \times 10^{-3} \times e^{0.96 \times \text{pH}}\right) - \frac{\text{F500nm}}{\text{F430nm}}} \quad (5)$$

Stability measurements of ArcC1 were performed as described above, with minor adjustments. In brief, ArcC1 was diluted in buffer B or 300 mM KPi, pH 7.0, with and without 200 mM glycine betaine, and incubated at 30 °C for 0, 1, 3 and 5 h. After incubation, 5 mM of ADP, 5 mM of MgSO$_4$ and either PercevalHR or pyranine were added. After 5 min of incubation at 30 °C, the reaction was started with the addition of 5 mM of carbamoyl-phosphate. For the experiments performed in 300 mM KPi pH 7.0 a new calibration of both PercevalHR and pyranine was performed, resulting in the following equation:

$$\frac{\text{ATP}}{\text{ADP}} = \frac{6.59 \times \left(\frac{\text{F500nm}}{\text{F430nm}} - \left(-0.214 + 8.2 \times 10^{-5} \times e^{1.29 \times \text{pH}}\right)\right)}{\left(-0.52 + 6.3 \times 10^{-4} \times e^{1.15 \times \text{pH}}\right) - \frac{\text{F500nm}}{\text{F430nm}}} \quad (6)$$

**Purification of ArcD2 and OpuA.** Membrane vesicles were quickly thawed and diluted to a total protein concentration of 3 mg mL$^{-1}$ for OpuA and 7 mg mL$^{-1}$ for ArcD2 in 50 mM KPi, pH 7.0 plus 200 mM KCl (buffer F) containing 20% (v/v) glycerol in case of OpuA. 0.5% (w/v) n-dodecyl-β-D-maltoside (DDM) was added to the vesicles for solubilization and the mixture was nutated for either 30 min (ArcD2) or 60 min (OpuA). Unsolubilized material was removed by centrifugation (20 min, 270,000 × g, 4 °C). Ni$^{2+}$-Sepharose resin (0.5 mL of Ni$^{2+}$-Sepharose resin per 20 mg total protein for OpuA or 0.25 mL of Ni$^{2+}$-Sepharose resin per 10 mg total protein for ArcD2) was pre-equilibrated in buffer F with 10 mM imidazole plus 0.03% (w/v) DDM. The supernatant was diluted 1.6× fold (ArcD2) or 2.5× fold (OpuA) to reduce the DDM concentration and then added to the Ni$^{2+}$-Sepharose column material and nutated for 1 h at 4 °C. The mixture was poured into a poly-prep column, after which the resin was washed with 20 column volumes of buffer F containing 50 mM imidazole plus 0.02% (w/v) DDM and 20% (v/v) glycerol in case of OpuA. Proteins were eluted in 3 column volumes of buffer F with 500 mM imidazole plus 0.02% (w/v) DDM and 20% (v/v) glycerol in case of OpuA.

**Light scattering for oligomeric state determination.** Ni$^{2+}$-Sepharose/size-exclusion chromatography-purified fractions of ArcA, ArcB, ArcC, and ArcD2 were analyzed on a second Superdex 200 Increase 10/300 GL size-exclusion column (GE Healthcare) in buffer D [with 0.02% (w/v) DDM for ArcD2], which was coupled to a multi-angle light scattering system with detectors for absorbance at 280 nm (Agilent Technologies, Inc.), static light scattering (Wyatt Technology Corporation) and differential refractive index (Wyatt Technology Corporation). Data analysis was performed with the ASTRA software package (Wyatt Technology Corporation), using a value for the refractive index increment (dn/dc)$_{protein}$ of 0.180 mL mg$^{-1}$ and (dn/dc)$_{detergent}$ of 0.143 mL mg$^{-1}$ [50].

**Co-reconstitution of ArcD2 and OpuA.** Synthetic lipids were mixed from chloroform stocks in the ratio of either 50 mol% DOPE, 12 mol% DOPC, and 38 mol% DOPG or 50 mol% DOPE, 37 mol% DOPC, and 13 mol% DOPG. Lipids were dried in a rotary vacuum setup (Büchi Labortechnik AG), dissolved in diethyl ether, dried again and rehydrated in buffer B to a final lipid concentration of 20 mg mL$^{-1}$. Dissolved lipids, cooled with ice water, were sonicated with a tip sonicator (Sonics and Materials, Inc.) (15 s on, 45 s off, 70% amplitude, 16 cycles), frozen-thawed three times, alternating between liquid nitrogen and (a water bath at) room temperature, and extruded 13 times through a 400 nm pore size polycarbonate filter (Avestin Europe GmbH) to obtain liposomes. Using an established protocol[41], ArcD2 and OpuA were co-reconstituted in preformed liposomes at a protein to lipid ratio of 1:2:400 (w/w), respectively. The liposomes were first diluted five times to a final concentration of 4 mg mL$^{-1}$ in buffer B with 25% (v/v) glycerol [final concentration 20% (v/v)] and then destabilized by a stepwise titration with 10% (v/v) Triton X-100, until the membrane was saturated with detergent (R$_{sat}$; ref. [51]), after which the membrane proteins were added. The purified proteins and destabilized liposomes were mixed for 15 min at 4 °C, after which detergent was removed by adding SM2 biobeads (600 mg per 20 mg of lipids) in three equal aliquots with 15 min incubation in between each addition. After the third addition,

**Table 2 Average number of molecules per vesicle**

| Compound | Internal concentration | Molecules per vesicle (84 nm radius) | Molecules per vesicle (225 nm radius) |
|---|---|---|---|
| Ornithine | 0.5 mM | 740 | 14,500 |
| ADP | 5 mM | 7400 | 144,600 |
| $Mg^{2+}$ | 5 mM | 7400 | 144,600 |
| Pyranine | 100 μM | 150 | 2900 |
| PercevalHR | 2.9 μM | 4 | 83 |
| Phosphate | 50 mM | 73,700 | 1,446,200 |
| $K^+$ | 50 mM | 73,700 | 1,446,200 |
| $Na^+$ | 25/40 mM | 36,800/59,000 | 723,100/1,157,000 |
| $Cl^-$ | 25/40 mM | 36,800/59,000 | 723,100/1,157,000 |
| ATP/ADP ratio of 3 (without GB uptake) | 3.75/1.25 mM | 5500/1800 | 108,500/36,200 |
| ATP/ADP ratio of 2 (with GB uptake) | 3.33/1.67 mM | 4900/2500 | 96,400/48,200 |
| Glycine betaine (imported) | 50 mM | 73,700 | 1,446,200 |

the mixture was incubated overnight, which was followed by one additional addition (200 mg per 20 mg of lipids) of SM2 biobeads and incubation for 1 h. Finally, the proteoliposomes were collected by centrifugation (2 h for 38% (w/w) DOPG lipids or 4 h for 13% (w/w) DOPG lipids, 125,000 × g, 4 °C) and resuspended in 200 μL per 20 mg of lipids, yielding a final lipid concentration of 100 mg mL$^{-1}$. For the $^{14}$C-arginine transport assay (see transport assays), reconstitution was done similarly as above, except that ArcD2 was reconstituted at a protein to lipid ratio of 2:400 (w/w). In addition, the proteoliposomes were diluted in buffer B without glycerol and centrifugation was done for 30 min, 325,000 × g, 4 °C.

Vesicles with a radius of 226 nm have $1.8 \times 10^6$ lipids per vesicle. Hence, a protein-to-lipid ratio of 400:1 (w/w) or $2.9 \times 10^4$ (mol/mol) for ArcD2 and 200:1 (w/w) or $5.7 \times 10^4$ (mol/mol) for OpuA yield 62 molecules of ArcD2 and 31 molecules of OpuA per vesicle.

**Encapsulation of the arginine breakdown pathway.** Protocol A1: The ArcD2- and OpuA-containing vesicles used for most studies were composed of 50 mol% DOPE, 12 mol% DOPC plus 38 mol% DOPG. The proteoliposomes (66 μL, 6.6 mg of lipid) containing ArcD2 and OpuA were mixed in buffer B with 1 μM ArcA, 2 μM ArcB, 5 μM ArcC1, 5 mM ADP, 5 mM MgSO$_4$, 0.5 mM ornithine and optionally 1.6–2.9 μM PercevalHR or 100 μM pyranine in a total volume of 200 μL; the final liposome concentration is 33 mg of lipid mL$^{-1}$. This yields a final buffer of 50 mM KPi pH 7.0 plus 25 mM NaCl (carried over with the purified ArcA, ArcB, and ArcC1) or 40 mM NaCl when PercevalHR is also included (see Table 2). The enzymes, metabolites and dyes were encapsulated by five freeze-thaw cycles in a 0.5 ml Eppendorf tube, alternating between liquid nitrogen and a 10 °C water bath, with vortexing of the tube before freezing. Next, the vesicles were extruded 13 times through a 400 nm pore size polycarbonate filter; the extruder was pre-washed in buffer A with 0.5 mM ornithine (buffer G). This procedure homogenizes the vesicles further and makes it likely that necessary components are present in all layers and compartments. The vesicles with encapsulated pyranine were then separated from free pyranine by running them over a 22 cm long Sephadex G-75 (Sigma) column in buffer G at 4 °C. To remove the residual external compounds, the vesicles were diluted to 6 mL in buffer G, collected by centrifugation (20 min, 325,000 × g, 4 °C) and washed with buffer G (6 mL), after which the vesicles were centrifuged and resuspended in 40 μL per 6.6 mg of lipid, yielding a final concentration of 165 mg of lipid mL$^{-1}$. Vesicles were kept on ice before subsequent measurements. Importantly, the ratio of the components inside the vesicles is very similar to the ratio in solution prior to encapsulation (Supplementary Fig. 6).

Protocol A2: Like protocol A1 except that the proteoliposomes were mixed with internal components in 60 mM KCl, yielding a final buffer of 15 mM KPi pH 7.0 plus 25 mM NaCl and 40 mM KCl.

Protocol A3: Like protocol A1 except that the vesicles were loaded with 15 mM ADP plus equimolar concentrations of MgSO$_4$.

Protocol A4: Like protocol A1 except that the vesicles were extruded through a 200 nm pore size polycarbonate filter.

Protocol A5: Like protocol A1 except that the vesicles were loaded with 10 mM ATP, 10 mM MgSO$_4$, 24 mM creatine-phosphate, and 2.4 mg mL$^{-1}$ creatine kinase in buffer B[40].

Protocol A6: Like protocol A1 except that the proteoliposomes containing ArcD2 and OpuA were first diluted in 6 ml of 50 mM NaPi, pH 7.0 (buffer H), collected by centrifugation (20 min, 325,000 × g, 4 °C) and resuspended in 66 μl of buffer H. They were then mixed in buffer H with the components mentioned above and the ArcA, ArcB, and ArcC1 purified in 50 mM NaPi instead of 50 mM KPi. In addition, the extruder was pre-washed and vesicles were resuspended in buffer H with L-ornithine. This encapsulation yields only sodium and no potassium ions inside the vesicles.

Protocol A7: Like protocol A1 except that the vesicles consisted of 50 mol% DOPE, 37 mol% DOPC plus 13 mol% DOPG.

**Cryo-EM analysis of vesicles.** The vesicles with encapsulated enzymes, metabolites (and sensors) were vitrified and imaging was done on a FEI Tecnai T20, 200 keV; Cryo-stage Gatan model 626. Samples were prepared under isosmotic conditions and images were recorded under low-dose conditions[52]. Approximate diameters of the vesicles were measured in ImageJ. The diameters were converted to internal volume by assuming spherical vesicles, multiplication by abundance and re-normalization.

**Transport assays.** Protocol B1: The vesicles with encapsulated enzymes and metabolites were diluted to a final concentration of 1.67 mg mL$^{-1}$ in buffer A with 250 mM KCl (buffer I). Glycine betaine was added at a final concentration of 18 μM, of which 2% (mol/mol) was $^{14}$C-radiolabeled. The mixture was incubated for 30 min at 30 °C. The internal ATP production was then started by addition of 20 mM arginine and samples of 50 μL were taken at given time intervals. Samples were diluted in 2 mL of ice-cold buffer I and filtered over 0.45 μm pore size cellulose nitrate filters to stop the transport assay. The filter was then washed with another 2 mL of buffer I. Radioactivity on the filter was quantified by liquid scintillation counting using Ultima Gold MV scintillation fluid (PerkinElmer) and a Tri-Carb 2800TR scintillation counter (PerkinElmer). The pore size of the filters is larger than the diameter of the vesicles, but the filters retain more than 99% of the vesicles and allow for rapid filtration[41].

Protocol B2: The transport of $^{14}$C-arginine was measured similarly, except that proteoliposomes (66 μL, 6.6 mg of lipid) with only ArcD2 in the membrane, and L-ornithine or L-citrulline (100 μM, 1 mM, or 10 mM) in buffer B in the vesicle lumen, were used (encapsulation was done by five freeze-thaw cycles in a total volume of 200 μL). The proteoliposomes were first extruded 13 times through a 400 nm pore size polycarbonate filter, then 13 times through a 200 nm filter and diluted to 6 mL in buffer B with or without the same concentration of L-ornithine or L-citrulline as on the inside. Proteoliposomes were collected by centrifugation (20 min, 225,000 × g, 4 °C) and either washed with buffer B (6 mL), centrifuged again and resuspended in 30 μL buffer B per 6.6 mg of lipid, or directly resuspended in buffer B with the appropriate concentration of L-ornithine or L-citrulline, yielding a final concentration of 220 mg of lipid mL$^{-1}$. For the transport assay, proteoliposomes were diluted to a final concentration of 2.2 mg of lipid mL$^{-1}$, in buffer B with 10 μM arginine, of which 10% (mol/mol) was $^{14}$C-radiolabeled, and 100 μL samples were taken at given time intervals. To impose a membrane potential, proteoliposomes in buffer B, were diluted 100-fold in 50 mM NaPi ($\Delta\Psi = -120$ mV), pH 7.0; 48.15 mM NaPi plus 1.85 mM KPi, pH 7.0 ($\Delta\Psi = -80$ mV); or 39.6 mM NaPi plus 10.4 mM KPi, pH 7.0 ($\Delta\Psi = -40$ mV), each supplemented with 1 μM of the potassium ionophore valinomycin.

**Internal ATP:ADP ratio measurements with PercevalHR.** Calibration: Purified PercevalHR and nucleotides (ATP and ADP) were encapsulated in liposomes. The encapsulation mixture contained liposomes at 7.5 mg of lipids mL$^{-1}$, 1.6–2.9 μM PercevalHR, 5 mM of nucleotides in varying ratios, and 5 mM MgSO$_4$ in buffer B. The samples were frozen-thawed five times, extruded 13 times through a 400 nm pore size polycarbonate filter, centrifuged twice (20 min, 225,000 × g, 4 °C) and finally resuspended in buffer B to a concentration of 167 mg mL$^{-1}$. The liposomes were diluted in buffer I with 10 μM carbonyl cyanide-4-(trifluoromethoxy) phenylhydrazone (FCCP) to a final concentration of 3.34 mg lipids mL$^{-1}$ in 105.250-QS cuvettes (Hellma Analytics) in a FP-8300 spectrofluorometer (Jasco, Inc.) and incubated for 3 min at 30 °C. The fluorescence spectrum of PercevalHR was measured by excitation from 400 ± 5 to 510 ± 5 nm, while the emission was recorded at 550 ± 5 nm. The encapsulated ATP/ADP ratio was plotted against the ratio of the peaks at 500 and 430 nm. Equation (3) was re-written with $n = 1$, to

obtain the following equation:

$$\frac{ATP}{ADP} = \frac{k * \left(\frac{F500nm}{F430nm} - start\right)}{end - \frac{F500nm}{F430nm}} \quad (7)$$

By fitting Eq. (7) to the data points (Supplementary Fig. 2d) the following values were obtained: $k = 3.02$, start $= 0.46$, end $= 1.30$.

Protocol B3: The vesicles with encapsulated enzymes, metabolites and PercevalHR were diluted in buffer I with 10 μM FCCP to a final concentration of 3.34 mg of lipid mL$^{-1}$ in 105.250-QS cuvettes (Hellma Analytics) in a FP-8300 spectrofluorometer (Jasco, Inc.) and incubated for 30 min at 30 °C. To start ATP production, 5 mM arginine was added, and after 30 min of incubation glycine betaine (0, 180 μM or 3.6 mM) was added. The fluorescence spectrum of PercevalHR was measured continuously, as described above. The ATP/ADP ratio was obtained from the ratio of the peaks at 500 and 430 nm, using Eq. (4). Representative traces are shown in the figure panels, but replicate experiments have been performed multiple times (*n*); *n* values are given in the figure legends.

**Internal pH measurements with pyranine.** Calibration: 100 μM pyranine was encapsulated in liposomes at varying pH between 6.0 and 8.0 in 50 mM KPi. The samples were frozen-thawed five times, extruded 13 times through a 400 nm pore size polycarbonate filter, and run over a 22 cm long Sephadex G-75 (Sigma) column. The vesicles were collected by centrifugation (20 min, 325,000 × *g*, 4 °C), washed once, centrifuged and resuspended to a concentration of 167 mg of lipid mL$^{-1}$. The liposomes were diluted in 50 mM KPi at varying pH to a final concentration of 3.34 mg lipids mL$^{-1}$ in 105.250-QS cuvettes (Hellma Analytics) in a FP-8300 spectrofluorometer (Jasco, Inc.) and incubated for 3 min at 30 °C. The fluorescence spectrum of pyranine was measured by excitation from 380 ± 5 to 480 ± 5 nm, while the emission was recorded at 512 ± 5 nm. The fluorescence spectrum of 0.1 μM pyranine in solution at varying pH between 6.0 and 9.0 in 50 mM KPi was also measured. The data points from encapsulated pyranine in unshocked vesicles overlapped perfectly with the in-solution data, therefore the pH in solution was plotted against the ratio of the peaks at 450 and 405 nm and fitted using the following logistic function:

$$y = \frac{L}{1 + e^{-k \times (x - x_0)}} \quad (8)$$

where *L* is the curve's maximum value; *k* is the logistic growth rate and $x_0$ is the *x*-value of the sigmoid's midpoint. Equation (8) was re-written with the following parameters: $x = $ pH and $y = $ F450nm/F405nm to obtain the following equation:

$$pH = \frac{\ln\left(L \times \frac{F405nm}{F450nm} - 1\right)}{-k} + x_0 \quad (9)$$

By fitting Eq. (8) to the data points in 50 mM KPi (Supplementary Fig. 4b), the following values were obtained: $L = 3.60$, $k = 2.38$, and $x_0 = 7.84$.

When vesicles are exposed to an osmotic upshift by the addition of 250 mM KCl, the internal KPi concentration increases to ~300 mM. The high salt concentration shifts the calibration of pyranine, and therefore additional calibration curves were made with pyranine in 300 mM KPi at pH values ranging from 6.0 and 9.0. This data was fitted to Eq. (8) (Supplementary Fig. 4b), as above, to obtain the following values: $L = 3.70$, $k = 2.21$, and $x_0 = 7.57$. The calibration is also shifted when the vesicles were encapsulated with 15 mM Mg-ADP instead of 5 mM Mg-ADP (see protocol A3). Therefore, we also made calibration curves with pyranine in 300 mM KPi plus 15 mM Mg-ADP at pH values ranging from 6.0 to 7.5 (with 15 Mg-ADP and 300 mM KPi the salts precipitated above pH 7.5). This data was fitted to Eq. (8) (Supplementary Fig. 4c) to obtain the following values: $L = 3.22$, $k = 2.16$, and $x_0 = 7.56$.

Protocol B4: The vesicles with encapsulated enzymes, metabolites and pyranine were diluted in buffer I to a final concentration of 3.34 mg lipids mL$^{-1}$ in 105.250-QS cuvettes (Hellma Analytics) in a FP-8300 spectrofluorometer (Jasco, Inc.) and incubated for 30 min at 30 °C. To start the ATP production, 5 or 20 mM arginine was added and after 30 min of incubation, glycine betaine was added at a concentration of 0 or 180 μM. The fluorescence spectrum of pyranine was measured as indicated above, and the internal pH was obtained from Eq. (9). Representative traces are shown in the figure panels, but replicate experiments have been performed multiple times (*n*); *n* values are given in the figure legends.

**External pH measurements with pyranine.** Calibration: The fluorescence spectrum of pyranine was measured by excitation from 380 ± 5 to 480 ± 5 nm, while the emission was recorded at 512 ± 5 nm. The fluorescence spectrum of 0.1 μM pyranine in solution at varying pH between 6.0 and 8.5 in 10 mM KPi plus 355 mM KCl was measured. The pH in solution was plotted against the ratio of the peaks at 450 and 405 nm and fitted using Eq. (8) (Supplementary Fig. 4d), from which the following values were obtained: $L = 3.27$, $k = 2.23$ and $x_0 = 7.48$.

Protocol B5: The vesicles with encapsulated enzymes and metabolites were diluted in 10 mM KPi pH 7.0 plus 355 mM KCl to a final concentration of 3.34 mg of lipids mL$^{-1}$ in 105.250-QS cuvettes (Hellma Analytics) in a FP-8300 spectrofluorometer (Jasco, Inc.) and incubated for 30 min at 30 °C. To start the ATP production, 5 mM arginine was added. The fluorescence spectrum of pyranine was measured as indicated above, and the internal pH was obtained

from Eq. (9). Representative traces are shown in the figure panels, but replicate experiments have been performed multiple times (*n*); *n* values are given in the figure legends.

**Amino acid and ammonia analysis.** Protocol B6: The vesicles with encapsulated enzymes and metabolites were diluted in buffer I, with or without glycine betaine, to a final concentration of 3.34 mg of lipid mL$^{-1}$ and incubated for 30 min at 30 °C. The assay was started by addition of 5 mM arginine and samples of 75 μL were taken at regular time intervals. Vesicles were removed from the samples by centrifugation (20 min, 225,000 × *g*, 4 °C) to stop the conversion of external amino acids. Subsequently, 50 μL of supernatant was mixed with 87.5 μL of 1 M boric acid, pH 9.0 (adjusted with 4 M KOH) and kept on ice before derivatization. In total, 37.5 μL of 99.8% methanol and 1.5 μL of 42 mM diethyl ethoxymethylenemalonate (DEEMM) were added to the samples for derivatization, after which the samples were incubated for 30 min in a sonication bath at room temperature, followed by 2 h of incubation in a heat block at 70 °C. The protocol for reverse-phase HPLC was adapted from ref. [53]. Amino acid samples were analyzed on a Shimadzu prominence HPLC system, containing a DGU-20A5R degassing unit, a LC-30AD solvent delivery unit, a SIL-30AC autosampler, a CTO-20AC column oven, and an SPD-M20A UV-VIS/Photodiode Array detector. A Shimadzu XR-ODS 3 × 75 mm C18 column was used to run the binary gradient with a flow rate of 0.9 mL min$^{-1}$ and an injection volume of 5 μL. Eluent A was 25 mM acetate, pH 5.8, supplemented with 0.02% (w/v) Na-azide. Eluent B was an 8:2 (v/v) mixture of acetonitrile and methanol. The gradient was as follows (all percentages are volume-based): start was at 94% Eluent A and 6% B; 87% A and 13% B at 2 min; 83% A and 17% B at 10 min; 71% A and 29% B at 11 min; 67% A and 33% B at 16 min; 40% A and 60% B at 16.1–18 min; 94% A and 6% B at 18.1 min toward the end of the protocol at 20 min. The compounds were identified based on retention times and quantified using the external standard method (Supplementary Fig. 7).

**Membrane permeability with stopped-flow fluorescence.** Principle of the method: To determine the permeability of solutes through the vesicle membrane, two independent and complementary fluorescence-based kinetic assays were used as described in M. Gabba et al. (manuscript in preparation). The first assay reports volume changes of vesicles by means of calcein self-quenching fluorescence[31]. The second assay monitors the pH variation in the vesicle lumen using the ratio-metric fluorophore pyranine[54]. Both assays exploit the out-of-equilibrium relaxation kinetics of the vesicles after the increase of the external osmotic pressure (osmotic upshift), i.e., the addition of a solute to the vesicle solution. After the osmotic upshift, the thermodynamic equilibrium is re-established by (i) water efflux (to re-equilibrate the chemical potential of water) and/or (ii) solute influx (to dissipate the solute concentration gradient). The contribution of the two fluxes to the recovery kinetics depends on the relative permeability of water and the solute as described in M. Gabba et al. (manuscript in preparation).

Osmolyte solutions: The 1 M stock solutions of Na-acetate, NH$_4$Cl, and NH$_4$-acetate were prepared by dissolving the salts into 100 mM KPi buffer; the pH was adjusted to 7.0 using 4 M NaOH. An empirical linear relation between concentration and osmolality for each solution was determined as described in M. Gabba et al. (manuscript in preparation). The stock solutions were then diluted to an osmolality of ca. 300 mosmol kg$^{-1}$ by mixing with the liposome solution (yielding 95 mM Na-acetate, 110 mM NH$_4$Cl, and 110 mM NH$_4$-acetate).

Liposome preparation: Liposomes were prepared as described under internal pH measurements with pyranine (see above), with minor adjustments. Calcein was added to the vesicle solution (2 mg of lipid in buffer A in a total volume of 1 mL) at a self-quenching concentration of 10 mM and enclosed by three cycles of freezing and thawing, alternating between liquid nitrogen and a 40 °C water bath. Pyranine (300 μM pyranine, final concentration) was encapsulated similarly. Next, the liposomes were extruded 13 times through a 200 nm pore size polycarbonate filter and run over a 22 cm long Sephadex G-75 (Sigma) column in buffer A. Vesicles were collected and diluted to a total volume of 12 mL in buffer A. Empty liposomes for blank correction were prepared using the same procedure without addition of calcein or pyranine.

Stopped-flow fluorescence measurements: The permeability of the liposomes for KPi, KCl, Na-acetate, NH$_4$Cl, and NH$_4$-acetate was assessed by monitoring the quenching of the fluorescence of calcein and by determining the changes in the internal pH using pyranine as a probe. A stopped-flow apparatus (SX20, Applied Photophysics Lim., Leatherhead, Surrey, UK) was used to measure fluorescence intensity kinetics upon imposition of an osmotic upshift to the liposomes filled with calcein or pyranine. The osmolyte (ca. 300 mosmol kg$^{-1}$ after mixing) and the liposome solutions (pre-equilibrated with 250 mM KCl) were loaded each in two distinct syringes, forced through the mixer (1:1 mixing ratio and 2 ms dead time) and into the optical cell (20 μl volume and 2 mm path length). The temperature was set at 20 °C using a water bath. The white light emitted by a Xenon arc lamp (150 W) was passed through a high precision monochromator and directed to the optical cell via an optical fiber. The band pass of the monochromator was optimized and set to 0.5 nm (for calcein) or 1.4 nm (for pyranine) to prevent fluorophore photobleaching during the experiment. The fluorophores were excited at 495 nm (for calcein) or at both 405 and 453 nm (for pyranine). The emitted light, collected at 90°, was filtered by a

Schott long-pass filter (cut-off wavelength at 515 nm) and detected by a photomultiplier tube (Hamamatsu R6095) with 10 µs time resolution. The voltage of the photomultiplier was automatically selected and kept constant during each set of experiments. The fluorescence intensity kinetics after the osmotic shock was recorded with logarithmically spaced time points to better resolve faster processes. For noise reduction, multiple acquisitions (three for slow kinetics and nine for fast kinetics) were performed for each experimental condition. Complete stopped-flow settings and acquired data processing are described in M. Gabba et al. (manuscript in preparation).

**Reporting summary**. Further information on research design is available in the Nature Research Reporting Summary linked to this article.

## Data availability

All data is available in the main text or supplementary materials. All data and materials used in the analysis are available upon request to the lead author.

## Code availability

All code used in the analysis is available upon reasonable request.

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

## Acknowledgements

We thank Giorgos Gouridis for the construction of the pNZarcA and pNZarcB vectors, Marc Stuart for assistance with the cryo-EM measurements, Cecile Deelman-Driessen for enzyme assays, Michiel Punter for assistance with data fitting, Dirk-Jan Slotboom and Juke Lolkema for discussions, and Wilhelm Huck and Ian Booth for critical reading of the paper. The work was funded by an ERC Advanced Grant (ABCvolume; #670578) and the Netherlands Organization for Scientific Research programs TOP-PUNT (#13.006) and Gravitation (BaSyC).

## Author contributions

B.G., B.P., H.S., T.P., and W.S. designed the research; B.G., H.S., T.P., and W.S. performed most of the research; S.S. purified enzymes; J.F. performed the stopped-flow measurements; B.G., B.P., H.S., T.P., and W.S. analyzed data; and B.G., B.P., H.S., and T.P. wrote the paper.

## Additional information

**Competing interests:** The authors declare no competing interests.

