## [Peer Review File · Nature Communications]

Reviewers' Comments:

Reviewer #1:

Remarks to the Author:

The authors reconstructed a synthetic metabolic network in which the physicochemical homeostasis can be maintained by the flux of arginine and glycine betaine. This reaction network has been reconstructed in liposome that has the size of few hundreds nanometer, thus mimicking a cell-like system. It is remarkable that two membrane proteins, the antiporter for arginine/ornithine and glycine betaine transporter. The authors analyzed well the kinetics of the reconstructed system based on the obtained data sets that look reliable.

Although the reconstructed system may be good for the detailed analysis of arginine metabolic pathway, the reviewer cannot find its evolvability and applicability for further study, especially in the study for entire construction of a living cell. For example, the authors claim "this study allows the development of complex cell-like systems with..." in the section of Conclusions. But any specific example has not been described.

Recent studies reconstituting cellular systems are trying more complex system, such as ATP photosynthesis for actin polymerization (Lee et al. 2018 Nat. Biotech.) or for protein synthesis (Berhanu et al. 2019 Nat. Commun.), Darwinian evolution in protocell (Ichihashi et al. 2013 nature Commun.), and phospholipid synthesis for membrane growing (Devaraj et al 2019 Nature Commun.). As compared to these achievements, the reviewer thinks this work does not have significant impact in the development of cell-like system or synthetic biology, therefore is not suited to be published in Nature Communications.

There are several incorrect and insufficient citations in the section of introduction as follows, these should be corrected and reconsidered.

P.2, line 20 ... biosynthesis lipid (8,9): Kuruma et al. BBA 1788 (2009) 567–574 should be additionally cited.

P.2, line 22 ... incorporated into vesicle (14,15): the authors cite incorrect paper. No. 14 is not using the recombinant cell-free system (13).

P.2, line 22 ... water-in-oil droplet (12): the authors cite incorrect paper. No. 12 is not using the recombinant cell-free system (13).

P.2, line 41 ... ATP dependent reactions (21): Berhanu et al. Nature Communications (2019) 10:1325 should be additionally cited.

Minor points

P.4, line 1 ... orientation of the protein.: this sentence should have a citation.

P.5, line 12 ... it is this that: this sentence should be rechecked.

P.6, line 23 (Fig. 3E; F): (Fig. 3E and F) ?

P.7, References and Notes, No. 4, DNA and RNA: Hachimoji DNA and RNA

The vesicle is composed of 13 or 38 % DOPG which has negative charge headgroup. On the other hand, positively charged arginine was added at the exterior of the vesicles. Therefore, the reviewer concerns aggregation of the vesicles that may destroy the compartment of the vesicle. Have authors ever observe the vesicle sample by microscopy or dynamic light scattering at this condition?

Reviewer #2:

Remarks to the Author:

I have read with interest and pleasure the quite original report presented by Prof. Poolman and collaborators. Let me say firstly that this is just a beautiful work and I surely support its publication in Nature Communications.

The work has relevant innovative elements when compared to similar research in the field of

bottom-up synthetic biology.

The science behind is very rigorous, and every step of the molecular system has been investigated in detail, so that the Authors can give a convincing explanation of all observations. All this is not improvised, but it clearly stems from years of experience.

There are several strong and new messages in this study. The first is the possibility of getting ATP not by the usual ATP synthase route, which requires a membrane and a previous process of proton motive force generation, but via substrate-driven phosphorylation. To the best of my knowledge this is the first time that researchers exploited this possibility in synthetic cell studies. It would be nice to read a small comment on this in the revised version, maybe highlighting alternative pathways to be explored in the future. This will contribute to the general progress of the field in next year.

The second message is that, in order to have a well-functioning network, the produced ATP must be used by another process. This is a profound message for all scholars involved in the field, that are forced to think in terms of out-of-equilibrium homeostasis, rather than monitoring single or complex reaction networks just running thermodynamically downhill. I think this is a key cue into the internal dynamics of living systems.

Some minor suggestions are:

- it is a pity not to find, in the reference list, any credit to pioneer people that have worked on synthetic cells – I mean here P.L. Luisi and T. Yomo. I do not believe that their contribution is so “minor” not to find space in the Authors’ reference list.
- Moreover, Luisi published one of the few other papers on homeostasis based on lipid synthesis and degradation – a topic that is conceptually similar to the central theme of this manuscript (Zepik et al., *Angew. Chem.* 2001, 40, 199)
- when the Authors comment on other systems for ATP production they correctly mention Lee et al. (*Nature Biotechnology*: ATP was used for actin polymerisation), but miss the opportunity to mention another important paper, published few months ago on *Nature Communication* (<https://www.nature.com/articles/s41467-019-09147-4>) by Kuruma, where ATP was used to fuel protein synthesis.
- Just a curiosity about the binding of ArcA: did the Authors attempted to tune the membrane composition in order to decrease ArcA binding to the external surface?
- More in general, did the authors perform a sort of optimisation of liposome membrane composition, and what was the result? Any interesting message for the readers?
- While reading the text (page 6), and Figure 3C, it is not very clear the comparison with the “creatine-phosphate/kinase system”: please explain it slightly better.
- Figure 1C is not immediately clear. Maybe it should be improved for the readers benefit.
- Another suggestion can be: Figure 1G, possibly, should come before Figure 1A, or even better, the pathway from arginine to ornithine (with reactants and enzymes explicitly mentioned) should be clearly drawn as Figure 1A.
- Is it possible to make a comparison between the amount of ATP produced by the present arginine-fuelled synthetic cells and the other two published systems (Lee et al., Berhanu et al.)?
- An aspect that is not sufficiently highlighted is the use of conventional (sub-micrometer) vesicles, whereas most of other studies are based on GVs. The Authors have shown that it is possible to

construct complex network inside conventional vesicles. The central point is the ability of incorporating membrane proteins, possibly in correct orientation. For OpuA this was achieved by previous knowledge, and for ArcD orientation did not matter. Do the Authors envision a general strategy for functionalising the membranes of conventional vesicles with membrane proteins in correct orientation? (so to progress synthetic cell research)

Reviewer #3:

Remarks to the Author:

I strongly believe that this manuscript is an interesting paper and containing novel advances in the bottom-up approaches in synthetic biology. Mainly metabolism process and rather clear explanation are implemented in their systems and further show its own homeostasis in artificially designed cell-like systems. The level and description of the experiments are very specific and show a high level of research. In specific, the detailed description and experimental evidences in Arginine breakdown are remarkable, showing a life-like homeostasis. Based on their experimental evidence and claim, I have no doubt that this paper must be published in Nature Communication. After reading the manuscript, I have several suggestions that the authors may consider in the revision if accepted.

1. Based on the experimental description, cryTEM has been performed. Although the size distribution and shapes were well characterized, but it will be necessary to show the actual images and analysis.
2. As described in P3, the vesicles are made of double or multi-layers. Then some interferences in the course of the reaction networks can be expected between layers. (Because the reconstituted proteins, such as ArcD and OpuA are probably existing each layer).
3. How the molecules (membrane proteins) per vesicle were estimated ? Any geometrical consideration due to the size of vesicles during the reconstitution process ?

REVIEWERS' COMMENTS:

Reviewer #1 (Remarks to the Author):

The authors reconstructed a synthetic metabolic network in which the physicochemical homeostasis can be maintained by the flux of arginine and glycine betaine. This reaction network has been reconstructed in liposome that has the size of few hundreds nanometer, thus mimicking a cell-like system. It is remarkable that two membrane proteins, the antiporter for arginine/ornithine and glycine betaine transporter. The authors analyzed well the kinetics of the reconstructed system based on the obtained data sets that look reliable.

Although the reconstructed system may be good for the detailed analysis of arginine metabolic pathway, the reviewer cannot find its evolvability and applicability for further study, especially in the study for entire construction of a living cell. For example, the authors claim “this study allows the development of complex cell-like systems with...” in the section of Conclusions. But any specific example has not been described. Recent studies reconstituting cellular systems are trying more complex system, such as ATP photosynthesis for actin polymerization (Lee et al. 2018 Nat. Biotech.) or for protein synthesis (Berhanu et al. 2019 Nat. Commun.), Darwinian evolution in protocell (Ichihashi et al. 2013 nature Commun.), and phospholipid synthesis for membrane growing (Devaraj et al 2019 Nature Commun.). As compared to these achievements, the reviewer thinks this work does not have significant impact in the development of cell-like system or synthetic biology, therefore is not suited to be published in Nature Communications.

REPLY: We disagree with the reviewer on the possibilities of evolvability and applicability of the system. We show volume regulation (e.g. ionic strength-gated import of betaine driven by the pathway for fuel) and physicochemical homeostasis, which is as complex as ATP photosynthesis or actin polymerization and crucial for the development of any cell-like system. Our system is different from published work and does not require a proton motive force as intermediate but synthesizes ATP directly. The importance of this achievement is emphasized by R2. The evolvability of the system requires production of the proteins from the corresponding genes, which is a long-term ambition that will require much more work (and beyond the scope of this study). However, the arginine breakdown pathway is ideal for the development of more complex cell-like system as the fuel system is readily combined with other ATP requiring processes, as e.g. emphasized by R2 and R3.

There are several incorrect and insufficient citations in the section of introduction as follows, these should be corrected and reconsidered.

REPLY: We apologize that some of the references got mixed up; the papers of Kuruma et al and Berhanu et al are cited in the revised manuscript.

P.2, line 20 ... biosynthesis lipid (8,9): Kuruma et al. BBA 1788 (2009) 567–574 should be additionally cited.

P.2, line 22 ... incorporated into vesicle (14,15): the authors cite incorrect paper. No. 14 is not using the

recombinant cell-free system (13).

P.2, line 22 ... water-in-oil droplet (12): the authors cite incorrect paper. No. 12 is not using the recombinant cell-free system (13).

P.2, line 41 ... ATP dependent reactions (21): Berhanu et al. Nature Communications (2019) 10:1325 should be additionally cited.

Minor points

P.4, line 1 ... orientation of the protein.: this sentence should have a citation.

P.5, line 12 ... it is this that: this sentence should be rechecked.

P.6, line 23 (Fig. 3E; F): (Fig. 3E and F) ?

P.7, References and Notes, No. 4, DNA and RNA: Hachimoji DNA and RNA

The vesicle is composed of 13 or 38 % DOPG which has negative charge headgroup. On the other hand, positively charged arginine was added at the exterior of the vesicles. Therefore, the reviewer concerns aggregation of the vesicles that may destroy the compartment of the vesicle. Have authors ever observe the vesicle sample by microscopy or dynamic light scattering at this condition?

REPLY: The minor points have been addressed. We have never observed any effect of arginine on the integrity of the vesicles. If arginine would destroy the compartment, we would not have observed the transport (accumulation of glycine betaine). In fact, we could monitor the transport for hours (e.g. Fig. 5c).

Reviewer #2 (Remarks to the Author):

I have read with interest and pleasure the quite original report presented by Prof. Poolman and collaborators. Let me say firstly that this is just a beautiful work and I surely support its publication in Nature Communications.

The work has relevant innovative elements when compared to similar research in the field of bottom-up synthetic biology.

The science behind is very rigorous, and every step of the molecular system has been investigated in detail, so that the Authors can give a convincing explanation of all observations. All this is not improvised, but it clearly stems from years of experience.

There are several strong and new messages in this study. The first is the possibility of getting ATP not by the usual ATP synthase route, which requires a membrane and a previous process of proton motive force generation, but via substrate-driven phosphorylation. To the best of my knowledge this is the first time that researchers exploited this possibility in synthetic cell studies. It would be nice to read a small comment on this in the revised version, maybe highlighting alternative pathways to be explored in the future. This will contribute to the general progress of the field in next year.

REPLY: In the final paragraph of the Results section (page 7) we now highlight alternative pathways that will be explored in the near future and cite a paper that we published recently.

The second message is that, in order have a well-functioning network, the produced ATP must be used by another process. This is a profound message for all scholars involved in the field, that are forced to think in terms of out-of-equilibrium homeostasis, rather than monitoring single or complex reaction networks just running thermodynamically downhill. I think this is a key cue into the internal dynamics of living systems.

REPLY: We thank the reviewer for his/her kind remarks and we agree with the statements on the new messages that we try to convey.

Some minor suggestions are:

- it is a pity not to find, in the reference list, any credit to pioneer people that have worked on synthetic cells – I mean here P.L.Luisi and T.Yomo. I do not believe that their contribution is so “minor” not to find space in the Authors’ reference list.

REPLY: We agree that the work of Luisi and Yomo is important and now cite some of their key papers (Luisi 2006 *Naturwissenschaften*, Ichihashi 2013 *Nat. Commun.* and Terasawa 2012 *Proc. Natl. Acad. Sci. U.S.A.*)

- Moreover, Luisi published one of the few other papers on homeostasis based on lipid synthesis and

degradation – a topic that is conceptually similar to the central theme of this manuscript (Zepik et al., *Angew. Chem.* 2001, 40, 199)

REPLY: This paper is now acknowledged.

- when the Authors comment on other systems for ATP production they correctly mention Lee et al. (*Nature Biotechnology*: ATP was used for actin polymerisation), but miss the opportunity to mention another important paper, published few months ago on *Nature Communication* (<https://www.nature.com/articles/s41467-019-09147-4>) by Kuruma, where ATP was used to fuel protein synthesis.

REPLY: Now included in the paper.

- Just a curiosity about the binding of ArcA: did the Authors attempted to tune the membrane composition in order to decrease ArcA binding to the external surface?

REPLY: We tested 38 and 13% DOPG and observed similar binding, but we always kept 50% DOPE for optimal activity of OpuA. We analyzed the binding in vesicles with and without ArcD2 and or OpuA and can rule out that ArcA is binding to either of the membrane proteins (page 4 of revised manuscript).

- More in general, did the authors perform a sort of optimisation of liposome membrane composition, and what was the result? Any interesting message for the readers?

REPLY: We have analyzed the lipid dependence of OpuA (described in ref. 24) and ArcD (unpublished) and used conditions that are compatible with ionic gating of OpuA and high activity of ArcD. We describe the pertinent lipid parameters on page 5. We aim to complete a more systematic study in the near future.

- While reading the text (page 6), and Figure 3C, it is not very clear the comparison with the “creatine-phosphate/kinase system”: please explain it slightly better.

REPLY: Rewritten as requested (page 6).

- Figure 1C is not immediately clear. Maybe it should be improved for the readers benefit.

REPLY: We have expanded the description in the legend of the new figure 1.

- Another suggestion can be: Figure 1G, possibly, should come before Figure 1A, or even better, the pathway from arginine to ornithine (with reactants and enzymes explicitly mentioned) should be clearly drawn as Figure 1A.

REPLY: We have reorganized the figure panels as also suggested by the editor; figure 1G now comes at the beginning.

- Is it possible to make a comparison between the amount of ATP produced by the present arginine-fuelled synthetic cells and the other two published systems (Lee et al., Berhanu et al.)?

REPLY: We can calculate ATP production rates, and we can increase them by incorporating more ArcD2 into the membrane and include more internal enzymes, but it is difficult to compare our ATP production rates with those of Lee et al. or Berhanu et al. We use ensembles of large vesicles (average diameter of 452 or 246 nm, according to volume), whereas Lee *et al* measure the synthesis of ATP in a few giant vesicles (10-100 μm). We report ATP/ADP ratios but since the total adenine nucleotide concentration is known, we can convert the data into molar values; e.g. ATP/ADP ratios of 2 to 3 (Fig. 5a, 6a) correspond to 3.33 to 3.75 mM of ATP, respectively, which is in the range of concentrations in living cells. The efficiency varies with the fraction of arginine converted into ornithine *versus* citrulline, which we can control by increasing the load on the system (through the uptake of glycine betaine, more ATP production and utilization).

- An aspect that is not sufficiently highlighted is the use of conventional (sub-micrometer) vesicles, whereas most of other studies are based on GVs. The Authors have shown that it is possible to construct complex network inside conventional vesicles. The central point is the ability of incorporating membrane proteins, possibly in correct orientation. For OpuA this was achieved by previous knowledge, and for ArcD orientation did not matter. Do the Authors envision a general strategy for functionalising the membranes of conventional vesicles with membrane proteins in correct orientation? (so to progress synthetic cell research)

REPLY: This is a very good point and not trivial to control with the present reconstitution technology. We do not consider a random orientation (observed for most membrane proteins) a real bottleneck as the fast majority of secondary transporters operate in both directions; for ATP driven transporters the orientation becomes problematic when the majority of the protein is inserted in the wrong direction but so far we have

not encountered this problem, neither with the proteins reconstituted here not with other transporters investigated in our laboratory. In the long run we will need to introduce the Sec translocation machinery for the unidirectional insertion of proteins into the membrane, but this is currently not possible.

Reviewer #3 (Remarks to the Author):

I strongly believe that this manuscript is an interesting paper and containing novel advances in the bottom-up approaches in synthetic biology. Mainly metabolism process and rather clear explanation are implemented in their systems and further show its own homeostasis in artificially designed cell-like systems. The level and description of the experiments are very specific and show a high level of research. In specific, the detailed description and experimental evidences in Arginine breakdown are remarkable, showing a life-like homeostasis. Based on their experimental evidence and claim, I have no doubt that this paper must be published in Nature Communication.

REPLY: We thank the reviewer for his/her comments.

After reading the manuscript, I have several suggestions that the authors may consider in the revision if accepted.

1. Based on the experimental description, cryTEM has been performed. Although the size distribution and shapes were well characterized, but it will be necessary to show the actual images and analysis.

REPLY: A representative micrograph is now shown for both filter sizes, exemplifying both the liposomes and the analysis. We cannot present all images as there are simply too many to show.

2. As described in P3, the vesicles are made of double or multi-layers. Then some interferences in the course of the reaction networks can be expected between layers. (Because the reconstituted proteins, such as ArcD and OpuA are probably existing each layer).

REPLY: We believe that the proteins after reconstitution indeed exist in each layer. Moreover, we believe that the soluble proteins and metabolites also end up in each compartment after encapsulation. Therefore, ATP can be produced in each layer. To verify that the ATP production is independent of the vesicle geometry, we have performed the experiments with vesicles extruded through a 200nm filter and a 400nm filter and have observed the same behavior (Figure 5c and Supplementary Figure 5).

3. How the molecules (membrane proteins) per vesicle were estimated ? Any geometrical consideration due to the size of vesicles during the reconstitution process ?

REPLY: The numbers of membrane proteins were estimated from the protein-to-lipid ratios (during the reconstitution). The number of lipids per vesicle is then derived using the average vesicle size from the EM analysis. We have added the following to the text (page 12): Vesicles with a radius of 226 nm have $1.8 \cdot 10^6$ lipids per vesicle. Hence, a protein-to-lipid ratio of 400:1 (w/w) or $2.9 \cdot 10^4$ (mol/mol) for ArcD2 and 200:1 (w/w) or $5.7 \cdot 10^4$ (mol/mol) for OpuA yield 62 molecules of ArcD2 and 31 molecules of OpuA per vesicle.